# Geometry and dynamics of representations in a precisely balanced memory network related to olfactory cortex

Claire Meissner-Bernard[1], Friedemann Zenke[1,2], Rainer W Friedrich[1,2]*

[1]Friedrich Miescher Institute for Biomedical Research, Basel, Switzerland; [2]University of Basel, Basel, Switzerland

*For correspondence:
rainer.friedrich@fmi.ch

Competing interest: The authors declare that no competing interests exist.

## eLife Assessment

This **important** study introduces a biologically constrained model of telencephalic area of adult zebrafish to highlight the significance of precisely balanced memory networks in olfactory processing. The authors provide **compelling** evidence that their model performs better in multiple situations (for e.g. in terms of network stability and shaping the geometry of representations), compared to traditional attractor networks and persistent activity. The work supports recent studies reporting functional E/I subnetworks in several sensory cortexes, and will be of interest to both theoretical and experimental neuroscientists studying network dynamics based on structured excitatory and inhibitory interactions.

**Abstract** Biological memory networks are thought to store information by experience-dependent changes in the synaptic connectivity between assemblies of neurons. Recent models suggest that these assemblies contain both excitatory and inhibitory neurons (E/I assemblies), resulting in co-tuning and precise balance of excitation and inhibition. To understand computational consequences of E/I assemblies under biologically realistic constraints we built a spiking network model based on experimental data from telencephalic area Dp of adult zebrafish, a precisely balanced recurrent network homologous to piriform cortex. We found that E/I assemblies stabilized firing rate distributions compared to networks with excitatory assemblies and global inhibition. Unlike classical memory models, networks with E/I assemblies did not show discrete attractor dynamics. Rather, responses to learned inputs were locally constrained onto manifolds that 'focused' activity into neuronal subspaces. The covariance structure of these manifolds supported pattern classification when information was retrieved from selected neuronal subsets. Networks with E/I assemblies therefore transformed the geometry of neuronal coding space, resulting in continuous representations that reflected both relatedness of inputs and an individual's experience. Such continuous representations enable fast pattern classification, can support continual learning, and may provide a basis for higher-order learning and cognitive computations.

## Introduction

Autoassociative memory establishes internal representations of specific inputs that may serve as a basis for higher brain functions including classification and prediction. Representation learning in autoassociative memory networks is thought to involve experience-dependent synaptic plasticity and potentially other mechanisms that enhance connectivity among assemblies of excitatory neurons

(*Hebb, 1949*; *Ko et al., 2011*; *Miehl et al., 2023*; *Ryan et al., 2015*). Classical theories proposed that assemblies define discrete attractor states and map related inputs onto a common stable output pattern. Hence, neuronal assemblies are thought to encode internal representations, or memories, that classify inputs relative to previous experience via attractor dynamics (*Amit and Tsodyks, 1991*; *Goldman-Rakic, 1995*; *Hopfield, 1982*; *Kohonen, 1984*; *Lagzi and Rotter, 2015*; *Mazzucato et al., 2015*). However, brain areas with memory functions such as the hippocampus or neocortex often exhibit dynamics that is atypical of attractor networks including irregular firing patterns, transient responses to inputs, and high trial-to-trial variability (*Iurilli and Datta, 2017*; *Renart et al., 2010*; *Shadlen and Newsome, 1994*).

Irregular, fluctuation-driven firing reminiscent of cortical activity emerges in recurrent networks when neurons receive strong excitatory (E) and inhibitory (I) synaptic input (*Brunel, 2000*; *Shadlen and Newsome, 1994*; *van Vreeswijk and Sompolinsky, 1996*). In such 'balanced state' networks, enhanced connectivity among assemblies of E neurons is prone to generate runaway activity unless matched I connectivity establishes co-tuning of E and I inputs in individual neurons. The resulting state of 'precise' synaptic balance stabilizes firing rates because inhomogeneities in excitation across the population or temporal variations in excitation are tracked by correlated inhibition (*Hennequin et al., 2017*; *Hennequin et al., 2014*; *Lagzi and Fairhall, 2024*; *Rost et al., 2018*; *Vogels et al., 2011*). E/I co-tuning has been observed experimentally in cortical brain areas (*Bhatia et al., 2019*; *Froemke et al., 2007*; *Okun and Lampl, 2008*; *Rupprecht and Friedrich, 2018*; *Wehr and Zador, 2003*) and emerged in simulations that included spike-timing-dependent plasticity at I synapses (*Lagzi et al., 2021*; *Litwin-Kumar and Doiron, 2014*; *Vogels et al., 2011*; *Zenke et al., 2015*). In simulations, E/I co-tuning can be established by including I neurons in assemblies, resulting in 'E/I assemblies' where I neurons track activity of E neurons (*Barron et al., 2017*; *Eckmann et al., 2024*; *Lagzi and Fairhall, 2024*; *Mackwood et al., 2021*). Exploring the structural basis of E/I co-tuning in biological networks is challenging because it requires the dense reconstruction of large neuronal circuits at synaptic resolution (*Friedrich and Wanner, 2021*).

Modeling studies started to investigate effects of E/I assemblies on network dynamics (*Chenkov et al., 2017*; *Mackwood et al., 2021*; *Sadeh and Clopath, 2020a*; *Schulz et al., 2021*) but the impact on neuronal computations in the brain remains unclear. Balanced state networks can exhibit a broad range of dynamical behaviors, including chaotic firing, transient responses, and stable states (*Festa et al., 2014*; *Hennequin et al., 2014*; *Litwin-Kumar and Doiron, 2012*; *Murphy and Miller, 2009*; *Roudi and Latham, 2007*), implying that computational consequences of E/I assemblies depend on network parameters. We therefore examined effects of E/I assemblies on autoassociative memory in a spiking network model that was constrained by experimental data from telencephalic area Dp of adult zebrafish, which is homologous to mammalian piriform cortex (*Mueller et al., 2011*).

Dp and piriform cortex receive direct input from mitral cells in the olfactory bulb (OB) and have been proposed to function as autoassociative memory networks (*Haberly, 2001*; *Wilson and Sullivan, 2011*). Consistent with this hypothesis, manipulations of neuronal activity in piriform cortex affected olfactory memory (*Meissner-Bernard et al., 2019*; *Sacco and Sacchetti, 2010*). In both brain areas, odors evoke temporally structured, spatially distributed activity patterns (*Blazing and Franks, 2020*; *Stettler and Axel, 2009*; *Yaksi et al., 2009*) that are dominated by synaptic inputs from recurrent connections (*Franks et al., 2011*; *Rupprecht and Friedrich, 2018*) and modified by experience (*Chapuis and Wilson, 2011*; *Frank et al., 2019*; *Jacobson et al., 2018*; *Pashkovski et al., 2020*). Whole-cell voltage clamp recordings revealed that neurons in posterior Dp (pDp) received large E and I synaptic inputs during odor responses. These inputs were co-tuned in odor space and correlated on fast timescales, demonstrating that pDp enters a transient state of precise synaptic balance during odor stimulation (*Rupprecht and Friedrich, 2018*).

We found that network models of pDp with assemblies but without E/I co-tuning generated persistent attractor dynamics and exhibited a biologically unrealistic broadening of the firing rate distribution. Introducing E/I assemblies established E/I co-tuning, stabilized the firing rate distribution, and abolished persistent attractor states. In networks with E/I assemblies, population activity was locally constrained onto manifolds that represented learned and related inputs by 'focusing' activity into neuronal subspaces. The covariance structure of manifolds supported pattern classification when information was retrieved from selected neuronal subsets. Furthermore, the continuity of the olfactory coding space provided a metric representing the similarity of inputs to learned stimuli. These results

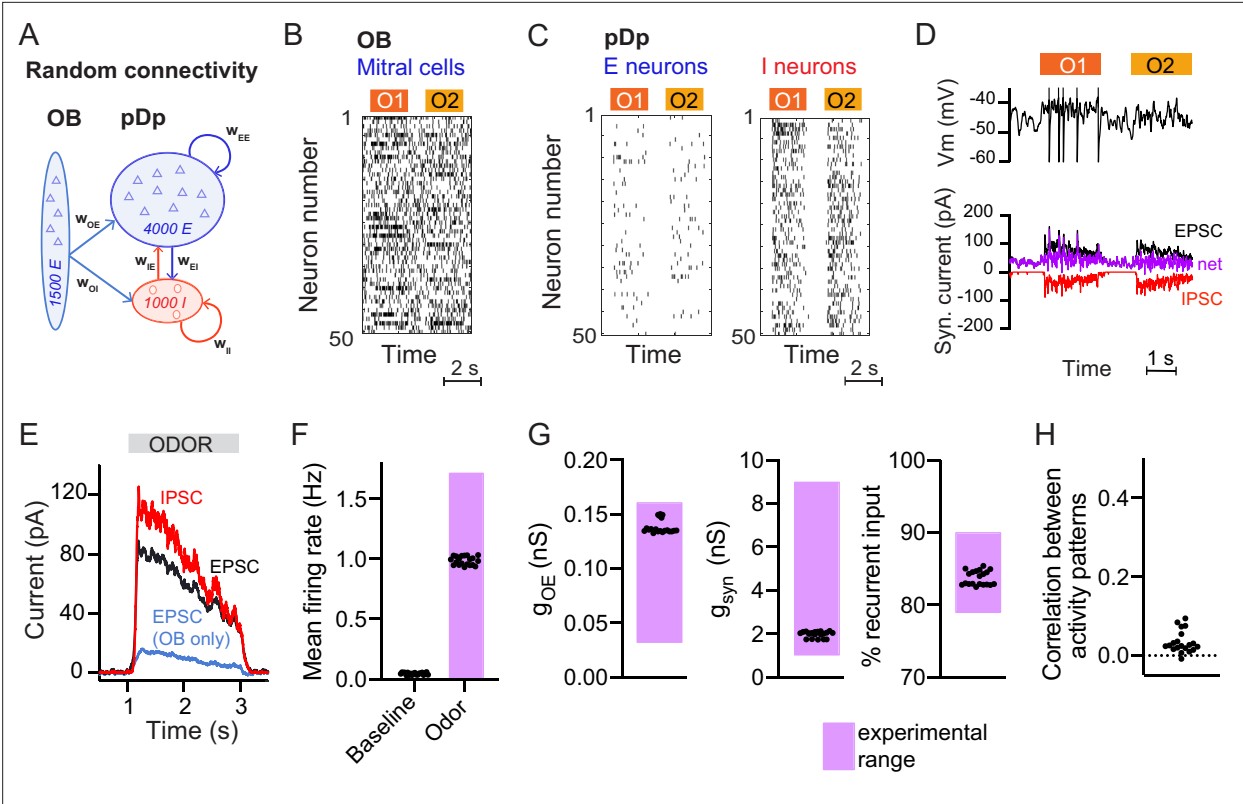

**Figure 1.** Spiking network model of pDp. (**A**) Schematic of pDp$_{sim}$. OB, olfactory bulb; E, excitatory; I, inhibitory neurons. (**B**) Spike raster of a random subset of 50 mitral cells in the OB representing 2 odors (O1 and O2). During odor stimulation, firing rates of 10% of mitral cells were increased and firing rates of 5% of mitral cells were decreased (baseline rate, 6 Hz). (**C**) Spike raster of random subsets of 50 E and I neurons in response to 2 odors. (**D**) Representative membrane potential trace (top) and excitatory (EPSC, black) and inhibitory (IPSC, red) currents (bottom) in one excitatory neuron in response to two odors. Purple trace shows net current (EPSC + IPSC). (**E**) Odor-evoked inhibitory (red) and excitatory (black and blue) currents as measured in a hypothetical voltage clamp experiment (conductance multiplied by 70 mV, the absolute difference between holding potential and reversal potential; *Rupprecht and Friedrich, 2018*). Representative example of one network, averaged across neurons and odors. (**F–H**) Measured values of the observables used to match pDp$_{sim}$ to experimental data. Each dot represents one network (average over 10 odors); $n$ = 20 networks. Pink shading shows the experimentally observed range of values. (**F**) Baseline and odor-evoked population firing rate. (**G**) Left: $g_{OE}$ is the synaptic conductance in E neurons contributed by afferents from the OB during odor stimulation. Middle: $g_{syn}$ is the total odor-evoked synaptic conductance. Right: % recurrent input quantifies the percentage of E input contributed by recurrent connections during odor stimulation. (**H**) Correlation coefficient between odor-evoked activity patterns in pDp$_{sim}$. The dotted line indicates the mean correlation between odor patterns in the OB.

show that autoassociative memory networks constrained by biological data operate in a balanced regime where information is contained in the geometry of neural manifolds. Predictions derived from these analyses may be tested experimentally by measurements of neuronal population activity in zebrafish.

## Results

### A spiking network model based on pDp

To analyze memory-related computational functions of E/I assemblies under biologically realistic constraints we built a spiking neural network model, pDp$_{sim}$, based on experimental data from pDp (*Figure 1A*; *Blumhagen et al., 2011*; *Rupprecht and Friedrich, 2018*; *Yaksi et al., 2009*). pDp$_{sim}$ comprised 4000 E neurons and 1000 I neurons, consistent with the estimated total number of 4000–10,000 neurons in adult zebrafish pDp (unpublished observations). The network received afferent input from 1500 mitral cells in the OB with a mean spontaneous firing rate of 6 Hz (*Friedrich and Laurent, 2004*; *Friedrich and Laurent, 2001*; *Tabor and Friedrich, 2008*). Odors were modeled by increasing the firing rates of 10% of mitral cells to a mean of 15 Hz and decreasing the firing rates of 5% of mitral cells to a mean of 2 Hz (Methods, *Figure 1B*, *Friedrich and Laurent, 2004*; *Friedrich and*

*Laurent, 2001*). As a result, the mean activity increased only slightly while the variance of firing rates across the mitral cell population increased approximately sevenfold, consistent with experimental observations (*Friedrich and Laurent, 2004*; *Wanner and Friedrich, 2020*).

pDp$_{sim}$ consisted of sparsely connected integrate-and-fire neurons with conductance-based synapses (connection probability ≤5%, Methods). Model parameters were taken from the literature when available and otherwise determined to reproduce measured observables (*Figure 1F–H*; Methods). The mean firing rate was <0.1 Hz in the absence of stimulation and increased to ~1 Hz during odor presentation (*Figure 1C, F*; *Blumhagen et al., 2011*; *Rupprecht et al., 2021*; *Rupprecht and Friedrich, 2018*). Population activity was odor-specific and activity patterns evoked by uncorrelated OB inputs remained uncorrelated in pDp$_{sim}$ (*Figure 1H*; *Yaksi et al., 2009*). The synaptic conductance during odor presentation substantially exceeded the resting conductance and inputs from other E neurons (recurrent inputs) contributed >80% of the excitatory synaptic conductance (*Figure 1G*). Hence, pDp$_{sim}$ entered a balanced state during odor stimulation (*Figure 1D, E*) with recurrent input dominating over afferent input, as observed in pDp (*Rupprecht and Friedrich, 2018*). Shuffling spike times of inhibitory neurons resulted in runaway activity with a probability of 0.79 ± 0.20, demonstrating that activity was inhibition-stabilized (*Sadeh and Clopath, 2020b*; *Tsodyks et al., 1997*). These results were robust against parameter variations (Methods). pDp$_{sim}$ therefore reproduced key features of pDp.

## Co-tuning and stability of networks with E/I assemblies

To create networks with defined neuronal assemblies we rewired a small subset of the connections in randomly connected (*rand*) networks. An assembly was generated by identifying the 100 E neurons that received the most connections from the mitral cells activated by a given odor and increasing the probability of connections between these E neurons by a factor of $\alpha$ (*Figure 2A* and *Figure 2—figure supplement 1A*). The number of E input connections per neuron was maintained by randomly eliminating connections from neurons outside the assembly. We thus refer to an odor as 'learned' when a network contains a corresponding assembly, and as 'novel' when no such assembly is present. In each network, we created 15 assemblies representing memories of uncorrelated odors. As a result, ~30% of E neurons were part of an assembly, with few neurons participating in multiple assemblies. Odor-evoked activity within assemblies was higher than the population mean and increased with $\alpha$ (*Figure 2B*). When $\alpha$ reached a critical value of ~6, networks became unstable and generated runaway activity (*Figure 2B*).

We first set $\alpha$ to 5, then uniformly scaled I-to-E connection weights by a factor of $\chi$ until E population firing rates in response to learned odors matched the corresponding firing rates in *rand* networks, that is, 1 Hz ('*Scaled I*' networks; *Figure 2A, C, D, F*). Under these conditions, activity within assemblies was amplified substantially in comparison to the corresponding neurons in *rand* networks (pseudo-assembly) whereas activity outside assemblies was substantially reduced (*Figure 2E, G*; *Figure 2—figure supplement 2*). Hence, non-specific scaling of inhibition resulted in a large and biologically unrealistic divergence of firing rates (*Figure 2—figure supplement 2*) that nearly exhausted the dynamic range of individual neurons in the population, indicating that homeostatic global inhibition is insufficient to maintain a stable firing rate distribution. We further observed that neurons within activated assemblies produced regular spike trains (*Figure 2—figure supplement 2*), indicating that the balanced regime was no longer maintained.

In *rand* networks, correlations between E and I synaptic conductances in individual neurons were slightly above zero (*Figure 2H*), presumably because of stochastic inhomogeneities in the connectivity (*Pehlevan and Sompolinsky, 2014*). In *Scaled I* networks, correlations remained near zero, indicating that E assemblies by themselves did not enhance E/I co-tuning (*Figure 2H*). *Scaled I* networks with structured E but random I connectivity can therefore not account for the stability, synaptic balance, and E/I co-tuning observed experimentally (*Rupprecht and Friedrich, 2018*).

We next created structured networks with more precise E/I balance by including I neurons within assemblies. We first selected the 25 I neurons that received the largest number of connections from the 100 E neurons of an assembly. The connectivity between these two sets of neurons was then enhanced by two procedures to generate E/I assemblies: (1) in '*Tuned I*' networks, the probability of I-to-E connections was increased by a factor $\beta$ while E-to-I connections remained unchanged. (2) In '*Tuned E+I*' networks, the probability of I-to-E connections was increased by $\beta$ and the probability of

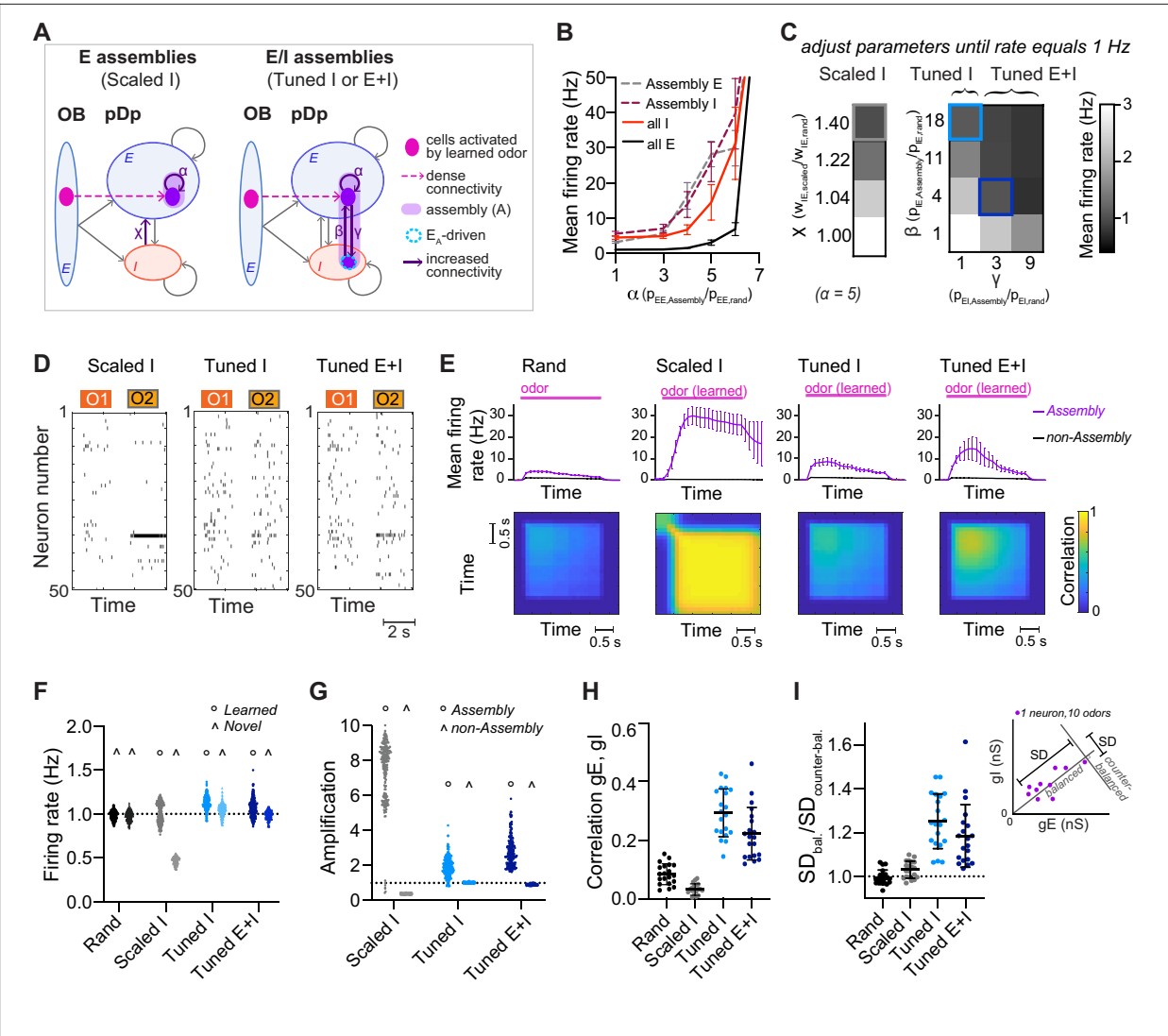

**Figure 2.** Networks with neuronal assemblies (memories). (**A**) Schematic of assemblies. Each assembly contains the 100 E neurons that are most densely connected to the mitral cells activated by a given odor. Connection probability between these E neurons is increased by a factor α. In *Scaled I* networks, weights of all I-to-E synapses are increased by a factor χ. In *Tuned* networks, the 25 I neurons that are most densely connected to the 100 E neurons are included in each assembly. In *Tuned I* networks, the probability of I-to-E connections within the assembly is increased by a factor β. In *Tuned E+I* networks, probabilities of I-to-E and E-to-I connectivity within the assembly are increased by factors β and γ, respectively. $n = 20$ networks with 15 assemblies each were simulated for each group. (**B**) Firing rates averaged over all E or I neurons (full lines) and over all assembly neurons (dashed lines) as a function of α (mean ± SD across 20 networks). (**C**) Mean E neuron firing rates of *Scaled* (left) and *Tuned* (right) networks in response to learned odors as a function of connection strength and probability, respectively. Squares depict parameters used in following figures unless stated otherwise. (**D**) Spike raster plots showing responses of 50 E neurons to 2 odors (O1: novel odor; O2: learned odor) in a *Scaled I* and the corresponding *Tuned* networks (same neurons and odors in the corresponding *rand* network are shown in **Figure 1C**). (**E**) Top: Mean firing rates in response to learned odors as a function of time, averaged across assembly or non-assembly neurons. Bottom: Mean correlation between activity patterns evoked by a learned odor at different time points during odor presentation. Correlation coefficients were calculated between pairs of activity vectors composed of the mean firing rates of E neurons in 100 ms time bins. Activity vectors were taken from the same or different trials, except for the diagonal, where only patterns from different trials were considered. The pink bar indicates odor presentation. (**F**) Mean firing rate in response to learned odors or novel odors. Each data point represents one network–odor pair ($n = 20$ networks, 10 odors). (**G**) Amplification within and outside assemblies, calculated as the ratio between the mean firing rates in response to learned odors averaged over the same populations of neurons in a given structured network (*Scaled I*, *Tuned I*, or *Tuned E+I*) and the corresponding *rand* network. Each data point represents one network–odor pair. (**H**) Quantification of co-tuning by the correlation between time-averaged E and I conductances in response to different odors, average across neurons. Each data point corresponds to one network ($n = 20$). Mean ± SD. (**I**) Quantification of co-tuning by the ratio of dispersion of joint conductances along balanced and counter-balanced axes (inset; Methods; $n = 20$ networks).

The online version of this article includes the following figure supplement(s) for figure 2:

*Figure 2 continued on next page*

*Figure 2 continued*

**Figure supplement 1.** Structured networks reproduce key features of pDp.

**Figure supplement 2.** Structured networks: additional results.

E-to-I connections was increased by $\gamma$ (*Figure 2A*; *Figure 2—figure supplement 1B*). As for '*Scaled I*' networks, $\beta$ and $\gamma$ were adjusted to obtain mean population firing rates of ~1 Hz in response to learned odors (*Figure 2C, D, F*). The other observables used to constrain the *rand* networks remained unaffected (*Figure 2—figure supplement 1C, D, E*). Activity of networks with E assemblies could not be stabilized around 1 Hz by increasing connectivity from subsets of I neurons receiving dense input from activated mitral cells (*Figure 2—figure supplement 1G, H*; *Sadeh and Clopath, 2020a*).

In *Tuned* networks, correlations between E and I conductances in individual neurons were significantly higher than in *rand* or *Scaled I* networks (*Figure 2H*). To further analyze E/I co-tuning we projected synaptic conductances of each neuron onto a line representing the E/I ratio expected in a balanced network (balanced axis) and onto an orthogonal line ('counter-balanced axis'; *Figure 2I*, inset, Methods). The ratio between the standard deviations (SDs) along these axes has been used previously to quantify E/I co-tuning in experimental studies (*Rupprecht and Friedrich, 2018*). This ratio was close to 1 in *rand* and *Scaled I* networks but significantly higher in *Tuned I* and *Tuned E+I* networks (*Figure 2I*). Hence, *Tuned* networks exhibited significant co-tuning along the balanced axis, as observed in pDp (*Rupprecht and Friedrich, 2018*).

In *Tuned* networks, activity within assemblies was higher than the mean activity but substantially lower and more irregular than in *Scaled I* networks (*Figure 2E, G*; *Figure 2—figure supplement 2*). Unlike in *Scaled I* networks, mean firing rates evoked by novel odors were indistinguishable from those evoked by learned odors and from mean firing rates in *rand* networks (*Figure 2F*, *Figure 2—figure supplement 1F*). Hence, E/I co-tuning prevented excessive amplification of activity in assemblies without affecting global network activity, consistent with experimental observations in Dp and piriform cortex (*Chapuis and Wilson, 2011*; *Frank et al., 2019*).

## Effects of E/I assemblies on attractor dynamics

We next explored effects of assemblies on network dynamics. In *rand* networks, firing rates increased after stimulus onset and rapidly returned to a low baseline after stimulus offset (*Figure 2E*). Correlations between activity patterns evoked by the same odor at different time points and in different trials were positive but substantially lower than unity, indicating high variability (*Figure 2E* and *Figure 2—figure supplement 2D*). Hence, *rand* networks showed transient and variable responses to input patterns, consistent with the typical behavior of generic balanced state networks (*Shadlen and Newsome, 1994*; *van Vreeswijk and Sompolinsky, 1996*). *Scaled I* networks responded to learned odors with persistent firing of assembly neurons and high pattern correlations across trials and time, implying attractor dynamics (*Hopfield, 1982*; *Khona and Fiete, 2022*), whereas *Tuned* networks exhibited transient responses and modest pattern correlations similar to *rand* networks (*Figure 2E*). Hence, *Tuned* networks did not exhibit stable attractor states, presumably because precise synaptic balance prevented strong recurrent amplification within E/I assemblies.

In classical memory networks, attractor dynamics drives pattern completion, the retrieval of the whole memory from noisy or corrupted versions of the learned input. We therefore tested whether networks with E/I assemblies performed pattern completion using two different approaches. In computational studies, pattern completion is often assessed by testing whether targeted partial stimulation of an assembly can activate the assembly as a whole (*Sadeh and Clopath, 2021*; *Vogels et al., 2011*). Stimulating subsets of E neurons in an assembly during baseline activity recruited the entire assembly in all structured pDp$_{sim}$ networks (*Scaled* and *Tuned*) without a significant increase in the overall population activity (*Figure 3—figure supplement 1A*).

During partial assembly stimulation, most pDp$_{sim}$ neurons do not receive any input, which is unlikely to happen during odor presentation. Therefore, we also assessed pattern completion with more realistic variations of inputs, building on experimental studies in the olfactory system and hippocampus (*Wilson and Sullivan, 2011*; *Yassa and Stark, 2011*). We morphed a novel odor into a learned odor (*Figure 3A*), or a learned odor into another learned odor (*Figure 3—figure supplement 1B*), and quantified the similarity between morphed and learned odors by the Pearson correlation of the OB activity patterns (input correlation). We then compared input correlations to the corresponding

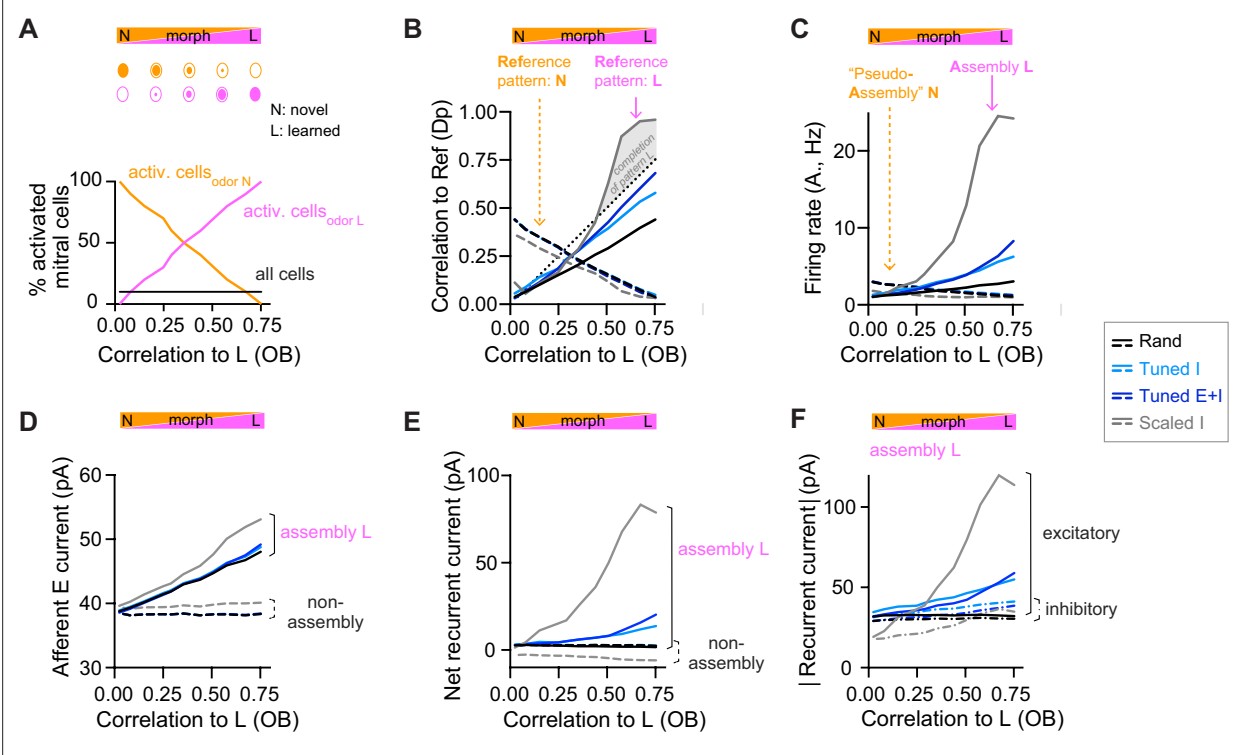

**Figure 3.** Changes of output activity to gradual modifications of inputs. (**A**) Morphing of a novel odor N into a learned odor L. Morphed odors were generated by gradually changing the fractions of activated mitral cells defining odors N or L. The x-axis indicates the similarity of a morphed odor to odor L, here quantified by the correlation between the olfactory bulb (OB) activity patterns representing the morphed and the learned odors (same x-axis in all panels). (**B**) Pearson correlation between activity patterns evoked by a morphed odor and a learned odor in pDp$_{sim}$ as a function of the corresponding correlations in the OB (full line). The dashed line shows the correlation to the novel odor instead of the learned odor. A signature of pattern completion is a steep increase in the correlation between activity patterns representing a morphed and a learned odor in pDp$_{sim}$, often exceeding the OB input correlation (gray shaded area). (**C**) Firing rates in response to morphed odors averaged across assembly neurons (learned odor, full line) or pseudo-assembly neurons (novel odor, dashed line) as a function of similarity between the presented odor and learned odor L (see A). (**D**) Excitatory input from the OB averaged over assembly neurons or non-assembly neurons. (**E**) Sum of inhibitory and excitatory currents evoked by other pDp$_{sim}$ neurons (recurrent input), averaged over assembly or non-assembly neurons. (**F**) Absolute inhibitory and excitatory recurrent currents averaged over assembly neurons. (**B–F**) Averages over eight networks.

The online version of this article includes the following figure supplement(s) for figure 3:

**Figure supplement 1.** Pattern completion: additional results.

pattern correlations among E neurons in pDp$_{sim}$ (output correlation). In *rand* networks, output correlations increased linearly with input correlations but did not exceed them (***Figure 3B***, ***Figure 3—figure supplement 1B***), consistent with the absence of pattern completion in generic random networks (***Babadi and Sompolinsky, 2014***; ***Marr, 1969***; ***Schaffer et al., 2018***; ***Wiechert et al., 2010***). *Scaled I* networks, in contrast, showed typical signatures of pattern completion: output correlations increased abruptly as a morphed odor approached a learned odor and eventually exceeded the corresponding input correlations (***Figure 3B***, ***Figure 3—figure supplement 1B***). In *Tuned* networks, output correlations were higher than in *rand* networks but remained lower than input correlations and did not increase steeply near the learned odor (***Figure 3B***, ***Figure 3—figure supplement 1B***). Hence, networks with E/I assemblies did not show signatures of pattern completion in response to naturalistic stimuli, consistent with the absence of strong recurrent amplification within assemblies.

To further understand the effect of E/I assemblies on network dynamics we examined the synaptic currents underlying the modest amplification of activity in assemblies of *Tuned* networks that occurred as a learned odor was approached (***Figure 3C***). We measured the relative contribution of synaptic inputs from afferent and recurrent sources received by assembly and non-assembly neurons. In *rand* networks, activity in pseudo-assembly neurons increased linearly as the corresponding odor was approached (***Figure 3C***). This linear increase in activity can be attributed entirely to an increase in

afferent input (*Figure 3D*) because the net recurrent input remained constant due to E/I balance (*Figure 3E*). In *Scaled* networks, assembly firing rates increased massively as the learned odor was approached (*Figure 3C*), reflecting a large increase in recurrent excitation (*Figure 3D–F*, *Figure 3— figure supplement 1C*). In *Tuned* networks, in contrast, assembly firing rates increased only modestly due to a slightly larger increase in E than I currents near the learned odor (*Figure 3D–F*, *Figure 3— figure supplement 1C*). Hence, the modest amplification by assemblies of *Tuned* networks involves an 'imperfect' synaptic balance.

## Geometry of activity patterns in networks with E/I assemblies

We next examined how E/I assemblies transform the geometry of neuronal representations, that is their organization in a state space where each axis represents the activity of one neuron or one pattern of neural covariance (*Chung and Abbott, 2021*; *Gallego et al., 2017*; *Langdon et al., 2023*). To address this general question, we created an odor subspace and examined its transformation by pDp$_{sim}$. The subspace consisted of a set of OB activity patterns representing four uncorrelated pure odors and mixtures of these pure odors. Pure odors were assigned to the corners of a square and mixtures were generated by selecting active mitral cells from each of the pure odors with probabilities depending on the relative distances from the corners (*Figure 4A*, Methods). Correlations between OB activity patterns representing pure odors and mixtures decreased approximately linearly as a function of distance in the subspace (*Figure 4B*). The odor subspace therefore represented a hypothetical olfactory environment with four odor sources at the corners of a square arena and concentration gradients within the arena. Locations in the odor subspace were visualized by the color code depicted in *Figure 4A*.

To examine how pDp$_{sim}$ transforms this odor subspace we projected time-averaged activity patterns onto the first two principal components (PCs). As expected, the distribution of OB activity patterns in PC space closely reflected the square geometry of the subspace (*Figure 4C*). This geometry was largely preserved in the output of *rand* networks, consistent with the notion that random networks tend to maintain similarity relationships between input patterns (*Figure 4D*; *Babadi and Sompolinsky, 2014*; *Marr, 1969*; *Schaffer et al., 2018*). We next examined outputs of *Scaled* or *Tuned* networks containing 15 assemblies, two of which were aligned with pure odors (*Figure 4—figure supplement 1A*). The four odors delineating the odor subspace therefore consisted of two learned and two novel odors. In *Scaled I* networks, odor inputs were discretely distributed between three locations in state space representing the two learned odors and residual odors, consistent with the expected discrete attractor states (*Figure 4D, E*). *Tuned* networks, in contrast, generated continuous representations of the odor subspace (*Figure 4D*). The geometry of these representations was distorted in the vicinity of learned odors, which were more distant from most mixtures than novel odors. These geometric transformations were less obvious when activity patterns of *Tuned* networks were projected onto the first two PCs extracted from *rand* networks (*Figure 4—figure supplement 1B*). Hence, E/I assemblies introduced local curvature into the coding space that partially separated learned from novel odors without disrupting the continuity of the subspace representation.

The curvature of the activity manifold in *Tuned* networks suggests that E/I assemblies confine population activity along specific dimensions of the state space. To test this hypothesis, we first quantified the dimensionality of odor-evoked activity by the participation ratio, a measure that depends on the eigenvalue distribution of the pairwise neural covariance matrix (*Altan et al., 2021*) (Methods). As expected, dimensionality was highest in *rand* networks and very low in *Scaled I* networks, reflecting the discrete attractor states (*Figure 4F*). In *Tuned* networks, dimensionality was high compared to *Scaled I* networks but lower than in *rand* networks (*Figure 4F*). The same trend was observed when we sampled data from a limited number of neurons to mimic experimental conditions and when we used other measures of dimensionality (*Figure 4—figure supplement 1C–E*). Furthermore, when restraining the analysis to activity evoked by novel odors and related mixtures, dimensionality was similar between *rand* and *Tuned* networks (*Figure 4F*). These observations support the hypothesis that E/I assemblies locally constrain neuronal activity onto lower-dimensional subspaces (manifolds), consistent with recent findings showing effects of specific circuit motifs on the dimensionality of neural activity (*Dahmen et al., 2023*; *Recanatesi et al., 2019*).

We further tested this hypothesis by examining the local geometry of activity patterns around representations of learned and novel odors. If E/I assemblies locally confine activity onto manifolds,

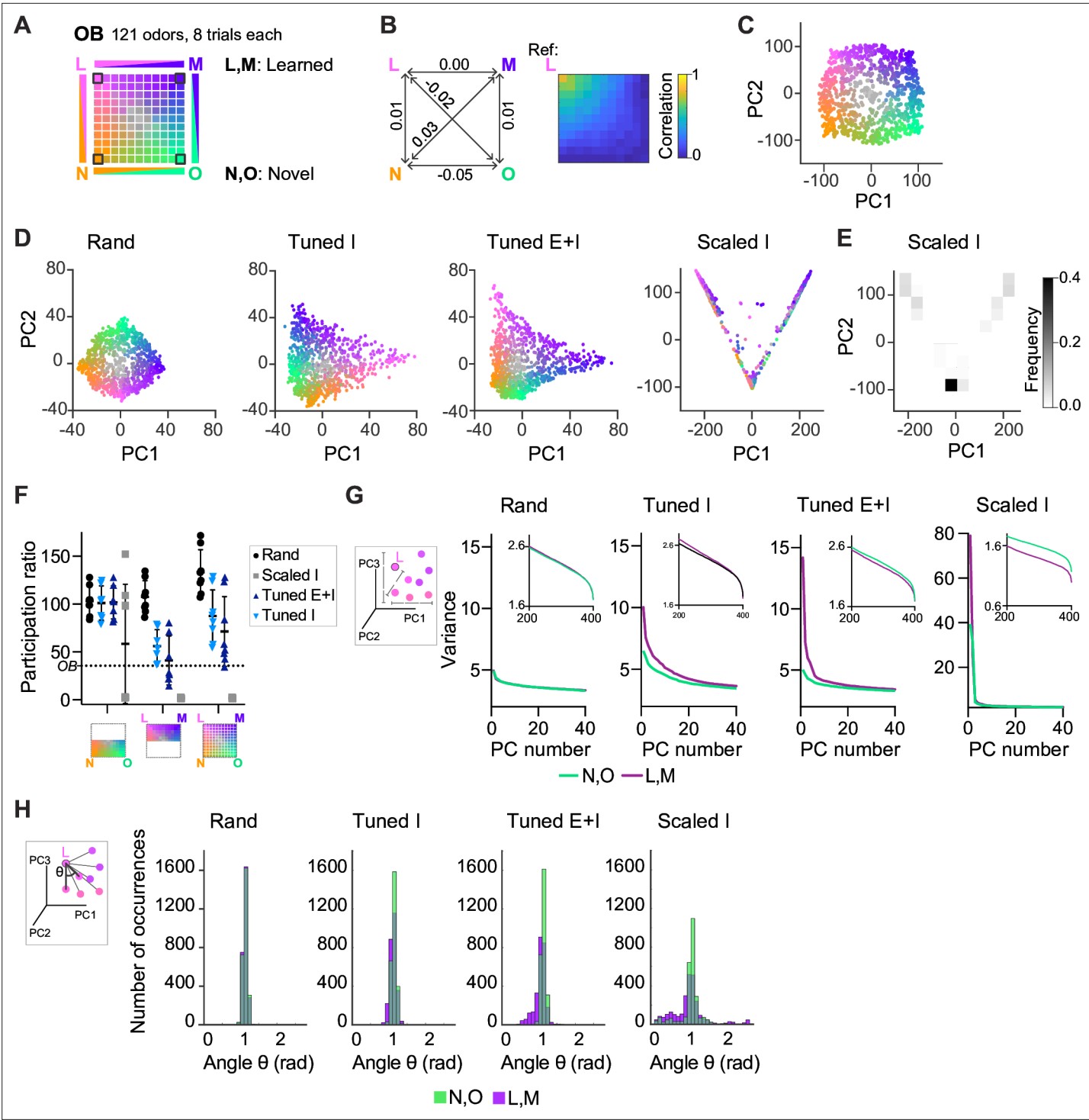

**Figure 4.** Geometry of odor representations in pDp$_{sim}$. (**A**) Odor subspace delineated by two learned (L, M) and two novel (N, O) pure odors at the vertices of a virtual square. Pixels within the square represent odor mixtures. (**B**) Left: Pearson correlations between olfactory bulb (OB) activity patterns defining pure odors. Right: Correlation between one pure odor (L; top left vertex) and all other odors. The odor from one vertex gradually morphs into the odor from another vertex. (**C**) Projection of activity patterns in the OB onto the first two principal components (PCs). Colors represent patterns in the odor subspace as shown in (**A, D**). Projection of activity patterns in pDp$_{sim}$ in response to the odor subspace onto the first two PCs. Representative examples of different pDp networks. (**E**) Density plot showing distribution of data points and demonstrating clustering at distinct locations in PC space for *Scaled I* networks. (**F**) Quantification of dimensionality of neural activity by the participation ratio: activity evoked by novel odors and related mixtures (left), activity evoked by learned odors and related mixtures (center), and activity evoked by all stimuli (right). Each data point represents one network (*n* = 8, mean ± SD); dotted line represents the participation ratio of OB activity. (**G**) Variance along the first 40 PCs extracted from activity patterns in

*Figure 4 continued on next page*

*Figure 4 continued*

different networks. Insets: Variance along PCs 200–400. (**H**) Angles between edges connecting a pure odor response and related versions thereof (inset). The analysis was performed using the first 400 PCs, which explained >75% of the variance in all networks. $n$ = 168 angles per pure odor in each of 8 networks (Methods). Similar results were obtained in the full-dimensional space.

The online version of this article includes the following figure supplement(s) for figure 4:

**Figure supplement 1.** Transformations and dimensionality of activity patterns: additional results.

**Figure supplement 2.** Geometry and dynamics in a simple model with equal parameters for excitation and inhibition.

**Figure supplement 3.** Manipulating amplification within assemblies.

small changes of input patterns should modify output patterns along preferred dimensions near representations of learned but not novel odors. To test this prediction, we selected sets of input patterns including each pure odor and the seven most closely related mixtures. We then quantified the variance of the projections of their corresponding output patterns onto the first 40 PCs (*Figure 4G*). In *Tuned* networks, this variance decreased only slightly as a function of PC rank for activity patterns related to novel odors, indicating that patterns varied similarly in all directions. For patterns related to learned odors, in contrast, the variance was substantially higher in the direction of the first few PCs, implying variation along preferred dimensions. In addition, we measured the distribution of angles between edges connecting activity patterns representing pure odors and their corresponding related mixtures in PC space (*Figure 4H*, inset; Methods; *Schoonover et al., 2021*). Angles were narrowly distributed around 1 rad in *rand* networks but smaller in the vicinity of learned patterns in *Tuned* networks (*Figure 4H*).

Activity may be constrained non-isotropically by amplification along a subset of dimensions, by inhibition along other dimensions, or both. E neurons participating in E/I assemblies had large loadings on the first two PCs (*Figure 4—figure supplement 1F, G*) and responded to learned odors with increased firing rates as compared to the mean rates in *Tuned E+I* and *rand* networks. Firing rates of the remaining neurons, in contrast, were slightly lower than the corresponding mean rates in *rand* networks. Consistent with these observations, the variance of activity projected onto the first few PCs was higher in *Tuned E+I* than in *rand* networks (*Figure 4G*) while the variance along higher-order PCs was lower (*Figure 4G*, inset). These observations indicate that local activity manifolds are delineated both by amplification of activity along preferred directions and by suppression of activity along other dimensions.

To obtain first insights into network mechanisms underlying effects of E/I assemblies on manifold geometry we simulated additional networks. Equalizing biophysical parameters of E and I neurons had little impact on network behavior (*Figure 4—figure supplement 2*; Methods), indicating that observed effects do not depend on specific biophysical differences between E and I neurons. We next explored the hypothesis that the modest amplification of activity within E/I assemblies contributes to the geometry of neural manifolds in pDp$_{sim}$. First, we adjusted the number of I-to-E connections ($\beta$) within assemblies of *Tuned E+I* networks to prevent amplification of activity within assemblies during presentation of learned odors (*Tuned[adjust]*, *Figure 4—figure supplement 3A–E*). In contrast to *Tuned E+I* networks with control parameters, we observed neither a curvature of activity manifolds in PC space nor an asymmetry of the Malalanobis distance ($d_M$; another signature of *Tuned* networks, see below) in these modified networks (*Figure 4—figure supplement 3F–H*). Second, we reduced recurrent amplification in assemblies of *Scaled I* networks by decreasing $\alpha$ and $\chi$ (*Scaled[adjust]*, *Figure 4—figure supplement 3A–E*). This resulted in distorted activity manifolds and an asymmetric increase in $d_M$, similar to observations in *Tuned* networks (*Figure 4—figure supplement 3F–H*; see below). Third, we artificially amplified activity in pseudo-assemblies of *rand* networks through concentration-dependent modifications of excitability instead of connectivity changes (*Figure 4—figure supplement 3A*). This manipulation increased the mean population activity as compared to control *rand* networks, but it also induced geometric modifications similar to *Tuned* networks, although the manifold curvature in PC space appeared less pronounced (*Figure 4—figure supplement 3B–H*). Hence, different manipulations had consistent effects on manifold geometry despite different effects on mean firing rates: signatures of manifold geometry observed in *Tuned* networks were mimicked by manipulations producing a modest, input-dependent amplification of firing rates in (pseudo-)assemblies but abolished by manipulations that suppressed this amplification. These results indicate that the modest

amplification of activity within assemblies contributes to the geometry of odor representations in pDp$_{sim}$. However, further analyses are required to explore potential contributions of other processes and to understand the underlying mechanisms in more detail.

## Pattern classification by networks with E/I assemblies

The lack of discrete attractor states raises the question how transformations of activity patterns by *Tuned* networks affect pattern classification. To quantify the association between an activity pattern and a class of patterns representing a given odor we computed the Mahalanobis distance ($d_M$). This measure quantifies the distance between the pattern and the class center, taking into account covariation of neuronal activity within the class. In bidirectional comparisons between patterns from different classes, the mean $d_M$ may be asymmetric if neural covariance differs between classes. We first quantified $d_M$ between representations of pure odors based on activity patterns across 80 E neurons drawn

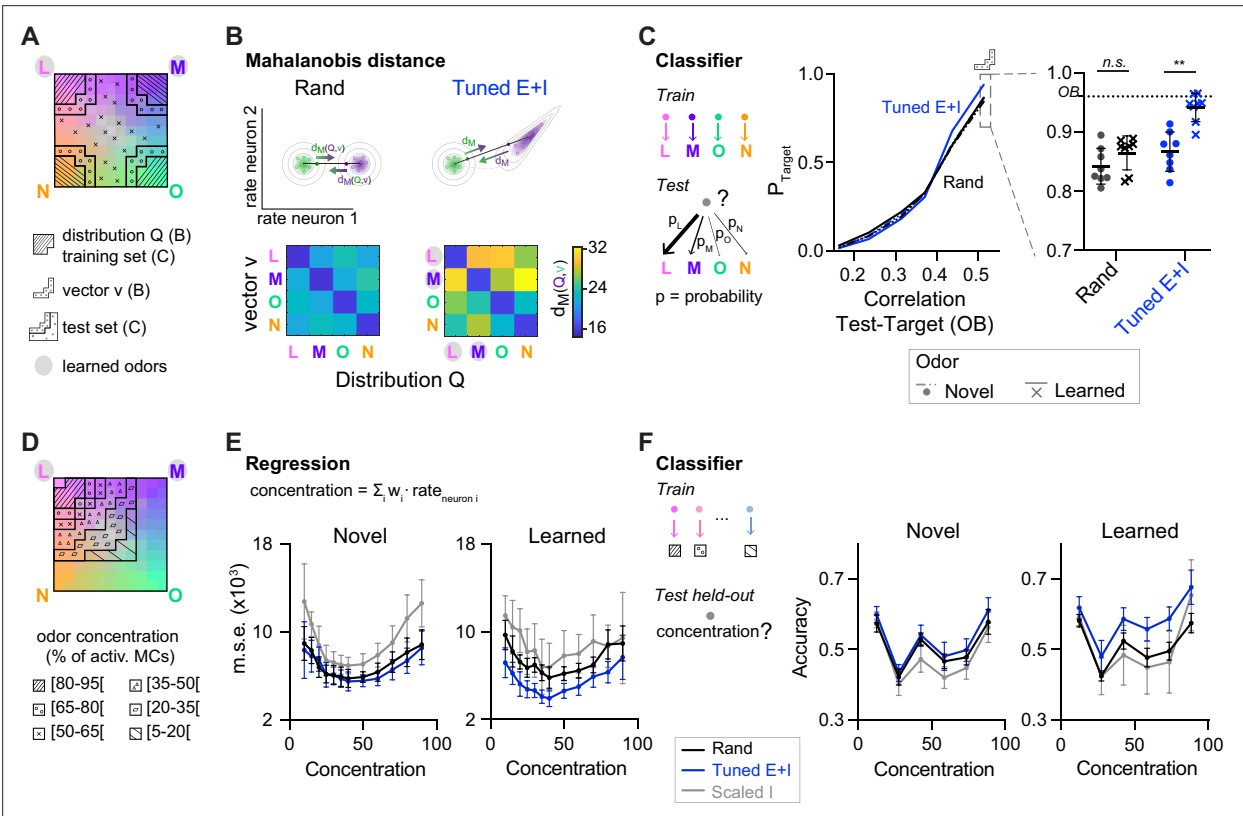

**Figure 5.** Distance relationships and classification of odor representations. (**A**) Odor-evoked activity patterns used as class distributions and vectors (**B**) or training and test sets (**C**). Same odor subspace as in *Figure 4*. (**B**) Top: Schematic illustration of Mahalanobis distance $d_M$. Bottom: $d_M$ between one activity vector (*v*) and reference classes (*Q*) in *rand* and *Tuned E+I* networks. $d_M$ was computed based on the activity patterns of 80 E neurons drawn from the four (pseudo-) assemblies. Average of 50 draws. Note that $d_M$ between patterns related to a learned odor and non-matching reference classes was higher in *Tuned E+I* networks. (**C**) Pattern classification probability quantified by quadratic discriminant analysis (QDA). $P_{Target}$ quantifies the probability that an activity pattern from the test set (odor mixtures, see **A**) is assigned to a target class from the training set (pure or closely related odor; see **A**). Left: Classification probability as a function of the similarity (Pearson correlation) between the test and target odors in the olfactory bulb (OB) (input patterns). Note enhanced classification probability for patterns evoked by odors similar to learned odors in *Tuned E+I* networks. Right: Classification probability for patterns similar to the training set (see **A**). Each data point represents one network (*n* = 8, mean ± SD; Wilcoxon signed-rank test, **: p < 0.01). (**D**) The odor subspace of *Figure 4* was subdivided in classes reflecting different concentrations of one of the pure odors (Methods). Odor concentration is defined here as the percentage of the 150 mitral cells representing a given odor that are activated. (**E**) The concentration of a given odor was regressed against the mean firing rates of a subset of neurons (Methods). Mean square error (squared difference between the actual concentration and the estimated concentration). (**F**) Accuracy of a linear support vector machine (SVM) in predicting the concentration of a novel (left) or learned odor (right) in a mixture. Each data point represents one network (*n* = 8, mean ± SD).

The online version of this article includes the following figure supplement(s) for figure 5:

**Figure supplement 1.** Further analyses of pattern distances: uncorrelated odors.

**Figure supplement 2.** Classification of odor representations (template matching).

from the corresponding (pseudo-) assemblies. $d_M$ was asymmetrically increased in *Tuned E+I* networks as compared to *rand* networks. Large increases were observed for distances between patterns related to learned odors and reference classes representing novel odors (*Figure 5A, B*). In the other direction, increases in $d_M$ were smaller. Moreover, distances between patterns related to novel odors were almost unchanged (*Figure 5B*). Further analyses showed that increases in $d_M$ in *Tuned E+I* networks involved both increases in the Euclidean distance between class centers and non-isotropic scaling of intra-class variability (*Figure 5—figure supplement 1*). The geometric transformation of odor representations by E/I assemblies therefore particularly enhanced the discriminability of patterns representing learned odors.

To further analyze pattern classification, we performed multi-class quadratic discriminant analysis (QDA), an extension of linear discriminant analysis for classes with unequal covariances. Using QDA, we determined the probabilities that an activity pattern evoked by a mixture is classified as being a member of each of four classes representing the pure odors, thus mimicking a four-way forced choice odor classification task. The four classes were defined by the distributions of activity patterns evoked by the pure and closely related odors. We then examined the classification probability of patterns evoked by mixtures with respect to a given class as a function of the similarity between the mixture and the corresponding pure odor (target) in the OB. As expected, the classification probability increased with similarity. Furthermore, in *Tuned E+I* networks, the classification probability of mixtures similar to a pure odor was significantly higher when the pure odor was learned (*Figure 5C*). Hence, E/I assemblies facilitated the classification of inputs related to learned patterns. Pattern classification by QDA could not be applied to outputs of *Scaled* networks because intra-class variability was too low (Methods).

E/I assemblies also improved pattern classification when activity patterns were classified based on correlations to template vectors representing the four odor classes (Methods; *Figure 5—figure supplement 2*). This analysis further revealed hysteresis effects in *Scaled* but not in *Tuned* networks that impaired pattern classification (*Figure 5—figure supplement 2*).

When neuronal subsets were randomly drawn not from assemblies but from the entire population, $d_M$ was generally lower (*Figure 5B*). These results indicate that assembly neurons convey higher-than-average information about learned odors. Together, these observations imply that pDp$_{sim}$ did not function as a categorical classifier but nonetheless supported the classification of learned odors, particularly when the readout focused on assembly neurons. Conceivably, classification may be further enhanced by optimizing the readout strategy, for example, by a learning-based approach. However, modeling biologically realistic readout mechanisms requires further experimental insights into the underlying circuitry.

## Additional computational properties of networks with E/I assemblies

E/I assemblies may not only support discrete classification but also other tasks. We hypothesized that continuous manifolds provide a distance metric to quantitatively evaluate the relatedness between an input and other stimuli. The geometric modifications induced by E/I assemblies may specifically provide a metric to evaluate distances between a new sensory input and previously learned stimuli. To explore these hypotheses, we first used multilinear regression to predict the concentration of a pure odor in subspace mixtures (*Figure 4A*) based on population activity. Highest prediction errors were observed for *Scaled I* networks, presumably because their discontinuous outputs poorly represent gradual concentration differences (*Figure 5D, E*). Lowest errors were observed for concentration estimates of learned odors in *Tuned* networks. We further trained support vector machines to estimate the concentrations of pure odors in the subspace mixtures and again found highest accuracy for learned odors in *Tuned* networks (*Figure 5F*). These results show that networks with E/I assemblies enable the quantitative analysis of selected stimulus components in the presence of distractors, presumably due to the geometric properties of the underlying neural manifold.

We next examined whether E/I assemblies can stabilize activity when multiple memories are successively added into a network, as in a continual learning process. In the absence of precise E/I balance, the divergence of firing rates and, thus, the risk of network instability are expected to increase as assemblies accumulate. *Tuned* networks may be more resistant against these risks, particularly when assemblies overlap, because activity is controlled more precisely within each additional assembly. To test this hypothesis, we examined responses of *Tuned E+I* and *Scaled I* networks to an additional odor

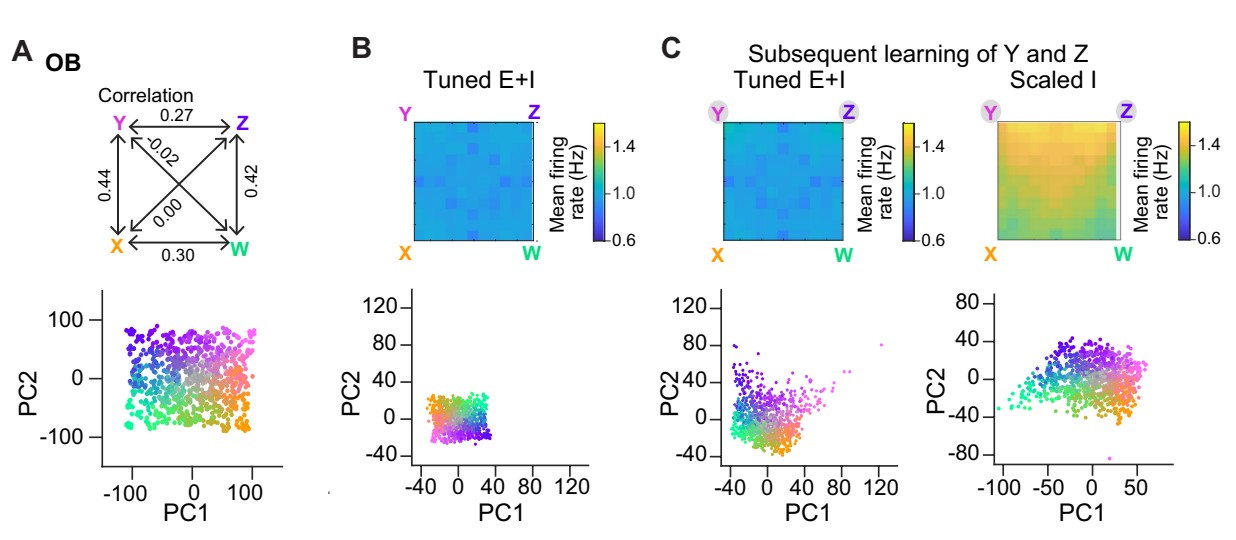

**Figure 6.** Representation of correlated patterns and resilience against additional memories. (**A**) Subspace delineated by four positively correlated odors (see Methods). Top: Correlations between pure odors. Bottom: Projection of olfactory bulb (OB) activity patterns onto the first two principal components (PCs). (**B**) Firing rates (top) and PC projection of output activity of a *Tuned E+I* network with 15 E/I assemblies that did not represent any of the four pure odors of the subspace. (**C**) Firing rates (top) and PC projection of output activity (bottom) after creation of two additional assemblies representing two of the pure odors (Y and Z). Left: *Tuned E+I* network. Right: *Scaled I* network. Note that in the *Scaled I* network, but not in the *Tuned E+I* network, firing rates evoked by newly learned odors were increased and patterns evoked by these odors were not well separated in PC space.

The online version of this article includes the following figure supplement(s) for figure 6:

**Figure supplement 1.** Further analyses of pattern distances: correlated odors.

subspace where four of the six pairwise correlations between the pure odors were clearly positive (range, 0.27–0.44; *Figure 6A*). We then compared networks with 15 randomly created assemblies to networks with two additional assemblies representing two of the correlated pure odors. These two additional assemblies had, on average, 16% of neurons in common due to the similarity of the odors. In *Tuned* networks, introducing additional overlapping assemblies selectively increased the discriminability of the corresponding odor representations in PC space (*Figure 6B, C*), as observed for uncorrelated assemblies (*Figure 4D*), but population firing rates remained almost unchanged (*Figure 6B, C*). In *Scaled I* networks, in contrast, creating two additional memories resulted in a convergence of the corresponding representations in PC space and in a substantial increase in firing rates, particularly in response to the learned and related odors (*Figure 6C*). These observations are consistent with the assumption that precise balance in E/I assemblies protects networks against instabilities during continual learning, even when memories overlap. We further observed that, in this regime of higher pattern similarity, $d_M$ was again increased upon learning, particularly between learned odors and reference classes representing other odors (*Figure 6—figure supplement 1*). E/I assemblies therefore consistently increased $d_M$ in a directional manner under different conditions.

## Discussion

### A precisely balanced memory network constrained by pDp

Autoassociative memory networks map inputs onto output patterns representing learned information. Classical models proposed this mapping to be accomplished by discrete attractor states that are defined by assemblies of E neurons and stabilized by global homeostatic inhibition. However, as seen in *Scaled I* networks, global inhibition is insufficient to maintain a stable, biologically plausible firing rate distribution. This problem can be overcome by including I neurons in assemblies, which leads to more precise synaptic balance. To explore network behavior in this regime under biologically relevant conditions we built a spiking network model constrained by experimental data from pDp. The resulting *Tuned* networks reproduced additional experimental observations that were not used as constraints including irregular firing patterns, lower output than input correlations, and the absence

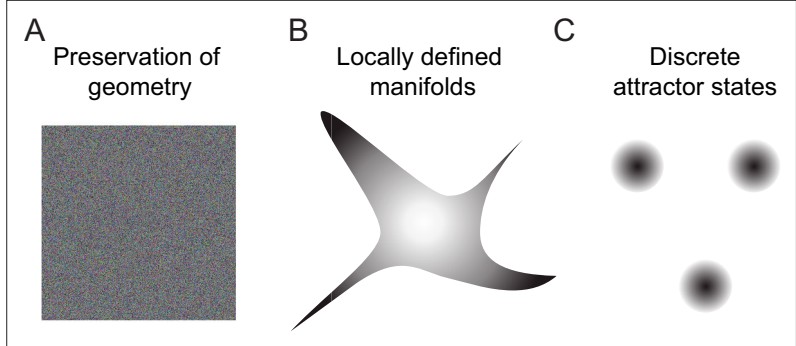

**Figure 7.** Schematic of geometric transformations. (**A**) Randomly connected networks tend to preserve the geometry of coding space. Such networks can support neuronal computations, for example, by projecting activity patterns in a higher-dimensional coding space for pattern classification. (**B**) We found that balanced networks with E/I assemblies transform the geometry of representations by locally restricting activity onto manifolds. These networks stored information about learned inputs while preserving continuity of the coding space. Such a geometry may support fast classification, continual learning and cognitive computations. Note that the true manifold geometry cannot be visualized appropriately in 2D because activity was 'focused' in different subsets of dimensions at different locations of coding space. As a consequence, the dimensionality of activity remained substantial. (**C**) Neuronal assemblies without precise balance established discrete attractor states, as observed in memory networks that store information as discrete items. Networks establishing locally defined activity manifolds (**B**) may thus be considered as intermediates between networks generating continuous representations without memories (**A**) and classical memory networks with discrete attractor dynamics (**C**).

of persistent activity. Hence, pDp$_{sim}$ recapitulated characteristic properties of a biological memory network with precise synaptic balance.

## Neuronal dynamics and representations in precisely balanced memory networks

Simulated networks with global inhibition showed attractor dynamics and pattern completion, consistent with classical attractor memory. However, the distribution of firing rates broadened as connection density within assemblies increased, resulting in unrealistically high (low) rates inside (outside) assemblies and, consequently, in a loss of synaptic balance. Hence, global inhibition was insufficient to stabilize population activity. In networks with E/I assemblies, in contrast, firing rates remained within a realistic range and the balanced state was maintained. Such *Tuned* networks showed no discrete attractor states but transformed the geometry of the coding space by confining activity to continuous manifolds near representations of learned inputs. This observation is consistent with the hypothesis that E/I co-tuning decreases the probability of multistable attractor states as compared to balanced networks with global inhibition or lateral inhibition between assemblies (*Boerlin et al., 2013*; *Hennequin et al., 2018*; *Wu and Zenke, 2021*).

Geometric transformations in *Tuned* networks may be considered as intermediate between two extremes: (1) geometry-preserving transformations as, for example, performed by many random networks (*Babadi and Sompolinsky, 2014*; *Marr, 1969*; *Schaffer et al., 2018*), and (2) discrete maps as, for example, generated by discrete attractor networks (*Freeman and Skarda, 1985*; *Hopfield, 1982*; *Khona and Fiete, 2022*; *Figure 7*). We found that transformations became more discrete map-like when amplification within assemblies was increased and precision of synaptic balance was reduced. Likewise, decreasing amplification in assemblies of *Scaled* networks changed transformations toward the intermediate behavior, albeit with broader firing rate distributions than in *Tuned* networks (*Figure 4—figure supplement 3B*). These observations indicate that a modest amplification of activity within assemblies contributes to geometric modifications of activity manifolds in *Tuned* networks, but other factors such as structured inhibition may also contribute. Hence, further analyses are required to obtain a deeper mechanistic understanding of manifold geometry in networks with E/I assemblies. Nonetheless, our results suggest that precise synaptic balance may generally favor intermediate over discrete transformations because this regime tends to linearize the relationship

between the mean input and output firing rates of neuronal populations (*Baker et al., 2020*; *Denève and Machens, 2016*).

E/I assemblies increased variability of population activity along preferred directions of state space and reduced its dimensionality in comparison to *rand* networks. Nonetheless, dimensionality remained high compared to *Scaled* networks with discrete attractor states. These observations indicate that geometric transformations in *Tuned* networks involved (1) a modest amplification of activity in one or a few directions aligned to the assembly, and (2) a modest reduction of activity in other directions. E/I assemblies therefore created a local curvature of coding space that 'focused' activity in a subset of dimensions and, thus, stored information in the geometry of coding space.

As E/I assemblies were small relative to the total size of the E neuron population, stored information may be represented predominantly by small neuronal subsets. Consistent with this hypothesis, $d_M$ was increased and the classification of learned inputs by QDA was enhanced when activity was read out from subsets of assembly neurons as compared to random neuronal subsets. Moreover, signatures of pattern completion were found in the activity of assemblies but not in global pattern correlations. The retrieval of information from networks with small E/I assemblies therefore depends on the selection of informative neurons for readout. Unlike in networks with global attractor states, signatures of memory storage may thus be difficult to detect experimentally without specific knowledge of assembly memberships.

## Computational functions of networks with E/I assemblies

In theory, precisely balanced networks with E/I assemblies may support pattern classification despite high variability of spike trains and the absence of discrete attractor states (*Denève and Machens, 2016*). Indeed, we found in *Tuned E+I networks* that input patterns were classified successfully by generic classifiers, particularly relative to learned inputs. Analyses based on the Mahalanobis distance $d_M$ indicate that classification of learned inputs was enhanced by two effects: (1) local manifolds representing learned odors became more distant from representations of other odors due to a modest increase in firing rates within E/I assemblies, and (2) the concomitant increase in variability was not isotropic, remaining sufficiently low in directions that separated novel from learned patterns. Hence, information contained in the geometry of coding space can be retrieved by readout mechanisms aligned to activity manifolds. Efficient readout mechanisms may thus integrate activity primarily from assembly neurons, as mimicked in our QDA-based pattern classification. This notion is consistent with the finding that the integrated activity of E/I assemblies can be highly informative despite variable firing of individual neurons (*Boerlin et al., 2013*; *Denève et al., 2017*; *Denève and Machens, 2016*). It will thus be interesting to explore how the readout of information from local manifolds could be further optimized.

Networks with E/I assemblies may also perform computations other than pattern classification. A toy model of continual learning indicates that precise balance can stabilize firing rate distributions when new memories are added, helping to prevent catastrophic failures. Furthermore, we found that continuous manifolds in *Tuned* networks support metric analyses, for example, to determine the relatedness of a (potentially complex) input to previously learned stimuli. Such tasks are not supported by discrete attractor states because information about gradual differences between inputs is lost by pattern completion. E/I assemblies can thus enhance the quantitative analysis of relevant information through changes in manifold geometry.

Conceivably, E/I assemblies may have further consequences for neuronal computation. Unlike discrete attractor networks, *Tuned* networks do not exhibit persistent activity, suggesting that they mediate fast computations rather than short-term memory functions. Fast classification may, for example, be important to interpret dynamical sensory inputs on a moment-to-moment basis. Moreover, the representation of learned inputs by small neuronal subsets, rather than global activity states, raises the possibility that multiple inputs can be classified simultaneously. Generally, the absence of discrete attractor states indicates that information is not stored in the form of distinct items. Rather, E/I assemblies cause geometric modifications of a continuous coding space that result in an overrepresentation of learned (relevant) inputs at the expense of other stimuli. These pattern transformations may thus contribute to different classification and learning processes by re-formatting and extracting relevant information for further processing by distributed networks. Such a function would be loosely related to computations within layers of artificial neuronal networks and consistent with the notion

that piriform cortex is embedded in a larger network comprising multiple telencephalic brain areas (*Haberly, 2001*).

## Balanced state networks with E/I assemblies as models for olfactory cortex

Piriform cortex and Dp have been proposed to function as attractor-based memory networks for odors. Consistent with this hypothesis, pattern completion and its modulation by learning has been observed in piriform cortex of rodents (*Barnes et al., 2008*; *Chapuis and Wilson, 2011*). However, odor-evoked firing patterns in piriform cortex and Dp are typically irregular, variable, transient and less reproducible than in the OB even after learning (*Jacobson et al., 2018*; *Pashkovski et al., 2020*; *Schoonover et al., 2021*; *Yaksi et al., 2009*), indicating that activity does not converge onto stable attractor states. Balanced networks with E/I assemblies, in contrast, are generally consistent with these experimental observations. Alternative models for pattern classification in the balanced state include networks endowed with short-term plasticity, which respond to stimuli with an initial amplification phase followed by a tonic inhibition-stabilized state (*Wu and Zenke, 2021*), or mechanisms related to 'balanced amplification', which typically generate pronounced activity transients (*Ahmadian and Miller, 2021*; *Murphy and Miller, 2009*). However, it has not been explored whether these models can be adapted to reproduce characteristic features of Dp or piriform cortex.

Our results generate predictions to test the hypothesis that E/I assemblies establish local manifolds in Dp: (1) odor-evoked population activity should be constrained onto manifolds, particularly in response to learned odors. (2) Learning should increase the magnitude and asymmetry of $d_M$ between odor representations. (3) Activity evoked by learned and related odors should exhibit lower dimensionality and more directional variability than activity evoked by novel odors. (4) Careful manipulations of inhibition may unmask assemblies by increasing amplification. These predictions may be addressed experimentally by large-scale measurements of odor-evoked activity after learning. The direct detection of E/I assemblies will ultimately require dense reconstructions of large neuronal networks at synaptic resolution. Given the small size of Dp, this challenge may be addressed in zebrafish by connectomics approaches based on volume electron microscopy (*Denk et al., 2012*; *Friedrich and Wanner, 2021*; *Kornfeld and Denk, 2018*).

The hypothesis that memory networks contain E/I assemblies and operate in a state of precise synaptic balance can be derived from the basic assumptions that (1) synaptic plasticity establishes assemblies and (2) that firing rate distributions remain stable as network structure is modified by experience (*Barron et al., 2017*; *Hennequin et al., 2017*). Hence, *Tuned* networks based on Dp may also reproduce features of other recurrently connected brain areas such as hippocampus and neocortex, which also operate in a balanced state (*Renart et al., 2010*; *Sadeh and Clopath, 2020b*; *Shadlen and Newsome, 1994*; *Znamenskiy et al., 2024*). Future experiments may therefore explore representations of learned information by local manifolds also in cortical brain areas.

## Methods

### Key resources table

| Reagent type (species) or resource | Designation | Source or reference | Identifiers | Additional information |
|---|---|---|---|---|
| Software, algorithm | Matlab | Mathworks | https://ch.mathworks.com (RRID:SCR_001622) | |

### pDp spiking network model

pDp$_{sim}$ consisted of 4000 excitatory (E) and 1000 inhibitory (I) neurons which were modeled as adaptive leaky integrate-and-fire units with conductance-based synapses of strength $w$.

A spike emitted at time $t$ by the presynaptic neuron $y$ from population $Y$ triggered an increase in the conductance $g_{Yx}$ in the postsynaptic neuron $x$, connected to $y$ with strength $w_{yx}$:

$$\tau_{\mathrm{syn},Y}\frac{dg_{Yx}}{dt} = -g_{Yx} + \tau_{\mathrm{syn},Y}\sum_y w_{yx}\delta\left(t - t_{\mathrm{spike},y}\right)$$

(1)

**Table 1.** Values of the neuronal parameters.
The superscripts indicate the reference where the experimental measurements can be found.

| Neuronal parameters | Symbol | Value | | |
|---|---|---|---|---|
| | | **Excitatory neuron** | **Inhibitory neuron** | **Simple model** |
| Membrane time constant | $\tau_X$ | 85 ms[*,†] | 50 ms[†] | 68 ms |
| Resting conductance | $g_{rest}$ | 1.35 nS[*] | 0.9 nS | 1.1 nS |
| Resting potential | $E_{rest}$ | −60 mV[*,†] | −65 mV | −62 mV |
| Excitatory reversal potential | $E_{exc}$ | 0 mV[*] | 0 mV | 0 mV |
| Inhibitory reversal potential | $E_{inh}$ | −70 mV[*] | −70 mV | −70 mV |
| Spiking threshold | $V_{th}$ | −38 mV[*] | −45 mV | −41 mV |
| Reset potential | $E_{rest}$ | −60 mV | −65 mV | −62 mV |
| Refractory period | $\tau_{ref}$ | 8 ms | 8 ms | 8 ms |
| Adaptation time constant | $\tau_a$ | 40 ms | / | 20 ms |
| Subthreshold adaptation | $a$ | 1 nS | / | 0.5 nS |
| Spike-triggered adaptation | $b$ | 10 pA | / | 5 pA |

[*]*Rupprecht and Friedrich, 2018*.
[†]*Blumhagen et al., 2011*.

The synaptic time constant $\tau_{syn,Y}$ was set to 10 ms for inhibitory synapses and 30 ms for excitatory synapses, except for the simple model were $\tau_{syn,Y}$ was 10 ms for both synapses.

Neuron $x$ from population $X$ received synaptic inputs from the OB as well as from the different local neuronal populations $P$. Its membrane potential $V_x$ evolved according to:

$$g_{rest,X}\, \tau_X \frac{dV_x}{dt} = g_{rest,X}\left(E_{rest,X} - V_x\right) + g_{OBx}\left(E_{exc} - V_x\right) + \sum_P g_{Px}\left(E_P - V_x\right) - z_x \qquad (2)$$

When the membrane potential reached a threshold $V_{th}$, the neuron emitted a spike and its membrane potential was reset to $E_{rest}$ and clamped to this value during a refractory period $\tau_{ref}$.

Excitatory neurons were endowed with adaptation with the following dynamics (*Brette and Gerstner, 2005*):

$$\tau_a \frac{dz}{dt} = a\left(V_x - E_{rest,E}\right) - z, \text{ with } z \to z + b \text{ after each spike} \qquad (3)$$

In inhibitory neurons, $z$ was set to 0 for simplicity.

The neuronal parameters of the model are summarized in *Table 1*. The values of the membrane time constant $\tau_X$, the resting conductance $g_{rest}$ and resting potential $E_{rest}$ of excitatory neurons, and the inhibitory and excitatory reversal potential $E_{inh}$ and $E_{exc}$ are in the range of experimentally measured

**Table 2.** Values of the connectivity parameters of different networks {probability $p_{YX}$ and synaptic strength $w_{YX}$ in pS}.
In *Figures 3–5*, 2 of the 5 networks from each structure were simulated. In the simple model, each neuron received the same number of inputs from each population.

| Network structure | OB → E | OB → I | E → E | E → I | I → E | I → I |
|---|---|---|---|---|---|---|
| A (n = 5) | {0.02, 128} | {0.01, 68} | {0.05, 128} | {0.04, 68} | {0.05, 480} | {0.04, 250} |
| B (n = 5) | {0.02, 128} | {0.01, 66} | {0.05, 128} | {0.04, 66} | {0.05, 450} | {0.04, 210} |
| C (n = 5) | {0.02, 128} | {0.01, 68} | {0.05, 108} | {0.02, 80} | {0.05, 520} | {0.02, 310} |
| D (n = 5) | {0.03, 95} | {0.02, 42} | {0.05, 128} | {0.04, 58} | {0.05, 590} | {0.04, 270} |
| Simple (n = 5) | {0.03, 160} | {0.03, 160} | {0.025, 370} | {0.025, 370} | {0.1, 1010} | {0.1, 1010} |

values (*Rupprecht and Friedrich, 2018*; *Blumhagen et al., 2011*). The remaining parameters were then fitted to fulfill two conditions (derived from unpublished experimental observations): (1) the neuron should not generate action potentials in response to a step current injection of duration 500 ms and amplitude 15 pA, and (2) the mean firing rate should be on the order of tens of Hz when the amplitude of the step current is 100 pA. Furthermore, the firing rates of inhibitory neurons should be higher than the firing rates of excitatory neurons, as observed experimentally (unpublished data).

To verify that the behavior of pDp$_{sim}$ was robust, we simulated 20 networks with different connection probabilities $p_{YX}$ and synaptic strengths $w_{YX}$ (*Table 2*). The connections between neurons were drawn from a Bernoulli distribution with a predefined $p_{YX} \leq 0.05$. As a consequence, each neuron received the same number of input connections. Care was also taken to ensure that the variation in the number of output connections across neurons was low in pDp$_{sim}$: connections of a presynaptic neuron $y$ to postsynaptic neurons $x$ were randomly deleted when their total number exceeded the average number of output connections by ≥5%, or added when they were lower by ≥5%.

The connection strengths $w_{YX}$ were then fitted to reproduce experimental observations in pDp (five observables in total, see below and *Figure 1*). For this purpose, a lower and an upper bound for $w_{YX}$ were set such that the amplitudes of single EPSPs and IPSPs were in the biologically plausible range of 0.2–2 mV. $w_{OE}$ was then further constrained to maintain the odor-evoked, time-averaged $g_{OE}$ in the range of experimental values (*Rupprecht and Friedrich, 2018*). Once $w_{OE}$ was fixed, the lower bound of $w_{EE}$ was increased to obtain a network rate >10 Hz in the absence of inhibition. A grid search was then used to refine the remaining $w_{YX}$.

## OB input

Each pDp$_{sim}$ neuron received external input from the OB, which consisted of 1500 mitral cells spontaneously active at 6 Hz. Odors were simulated by increasing the firing rate of 150 randomly selected mitral cells. Firing rates of these 'activated' mitral cells were drawn from a discrete uniform distribution ranging from 8 to 32 Hz and their onset latencies were drawn from a discrete uniform distribution ranging from 0 to 200 ms. An additional 75 randomly selected mitral cells were inhibited. Firing rates and latencies of these neurons were drawn from discrete uniform distributions ranging from 0 to 5 Hz and from 0 to 200 ms, respectively. After odor onset, firing rates decreased exponentially with a time constant of 1, 2, or 4 s (equally distributed). Spikes were then generated from a Poisson distribution, and this process was repeated to create trial-to-trial variability. As all odors had almost identical firing patterns, the total OB input did not vary much across odors. In *Figures 1–3*, the odor set consisted of 10 novel and/or 10 learned odors, all of which were uncorrelated (pattern correlations close to zero). Odors were presented for 2 s and separated by 1 s of baseline activity.

**Table 3.** Percentage of mitral cells out of the 150 mitral cells defining one pure odor (top left corner) that are activated in mixtures.
Percentages may be interpreted as relative odor concentration. Values given here are for uncorrelated pure odors.

| | | | | | | | | | | |
|-----|-----|-----|-----|-----|-----|-----|-----|-----|-----|-----|
| 100 | 90 | 80 | 70 | 60 | 50 | 40 | 30 | 20 | 10 | 0 |
| 90 | 90 | 80 | 70 | 60 | 50 | 40 | 30 | 15 | 5 | 0 |
| 80 | 80 | 70 | 60 | 50 | 40 | 35 | 25 | 15 | 5 | 0 |
| 70 | 70 | 60 | 50 | 50 | 35 | 30 | 20 | 15 | 5 | 0 |
| 60 | 60 | 50 | 50 | 40 | 30 | 25 | 20 | 15 | 5 | 0 |
| 50 | 50 | 40 | 35 | 30 | 25 | 25 | 15 | 10 | 5 | 0 |
| 40 | 40 | 35 | 30 | 25 | 25 | 15 | 10 | 10 | 5 | 0 |
| 30 | 30 | 25 | 20 | 20 | 15 | 10 | 10 | 5 | 5 | 0 |
| 20 | 15 | 15 | 15 | 15 | 10 | 10 | 5 | 5 | 5 | 0 |
| 10 | 5 | 5 | 5 | 5 | 5 | 5 | 5 | 5 | 0 | 0 |
| 0 | 0 | 0 | 0 | 0 | 0 | 0 | 0 | 0 | 0 | 0 |

**Table 4.** Percentage of activated cells available for selection for uncorrelated pure odors. C is obtained by multiplying the values by 1.5.

| | | | | | | | | | | |
|---|---|---|---|---|---|---|---|---|---|---|
| 100 | 100 | 100 | 90 | 80 | 70 | 60 | 50 | 40 | 30 | 0 |
| 100 | 100 | 100 | 90 | 80 | 70 | 60 | 50 | 35 | 25 | 0 |
| 100 | 100 | 90 | 60 | 70 | 60 | 55 | 45 | 35 | 25 | 0 |
| 90 | 90 | 80 | 70 | 70 | 55 | 50 | 40 | 35 | 25 | 0 |
| 80 | 80 | 70 | 70 | 60 | 50 | 45 | 40 | 35 | 25 | 0 |
| 70 | 70 | 60 | 55 | 50 | 45 | 45 | 35 | 30 | 25 | 0 |
| 60 | 60 | 55 | 50 | 45 | 45 | 35 | 30 | 30 | 25 | 0 |
| 50 | 50 | 45 | 40 | 40 | 35 | 30 | 30 | 25 | 25 | 0 |
| 40 | 35 | 35 | 35 | 35 | 30 | 30 | 25 | 25 | 25 | 0 |
| 30 | 25 | 25 | 25 | 25 | 25 | 25 | 25 | 25 | 0 | 0 |
| 0 | 0 | 0 | 0 | 0 | 0 | 0 | 0 | 0 | 0 | 0 |

Olfactory subspaces comprised 121 OB activity patterns. Each pattern was represented by a pixel in a 11 × 11 square. The pixel at each vertex corresponded to one pure odor with 150 activated and 75 inhibited mitral cells as described above, and the remaining pixels corresponded to mixtures. When pure odors were correlated (*Figure 6*), adjacent pure odors shared half of their activated and half of their inhibited mitral cells. We generated eight different trajectories within the square, each visiting all possible virtual odor locations for 1 s. Each trajectory thus consisted of 121 s of odor presentation, and trajectories were separated by 2 s of baseline activity. The dataset for analysis therefore comprised 968 activity patterns (8 trajectories × 121 odors). The total number of activated and inhibited cells at each location in the virtual square remained within the range of 150 ± 10% and 75 ± 10%, respectively. The fraction of activated and inhibited mitral cells from a given pure odor (its concentration) decreased with the distance from the corresponding vertex as shown in *Table 3*. At each location within the square and for each trajectory, mitral cells that remained activated in the mixture were randomly selected from the pool of C mitral cells with the shortest latencies from each pure odor. C decreased with the distance from the vertex as shown in *Table 4*. The firing rate of each selected mitral cell varied ±1 Hz around its rate in response to the pure odor. The identity, but not the firing rate, of the activated mitral cells therefore changed gradually within the odor subspace. This procedure reflects the experimental observation that responses of zebrafish mitral cells to binary odor mixtures often resemble responses to one of the pure components (*Tabor et al., 2004*).

## Assemblies

Unless noted otherwise, *Scaled* and *Tuned* networks contained 15 assembles (memories). An assembly representing a given odor contained the 100 E neurons that received the highest density of inputs from the corresponding active mitral cells. Hence, the size of assemblies was substantially smaller than the total population, consistent with the observation that only a minority of neurons in Dp or piriform cortex are activated during odor stimulation (*Miura et al., 2012*; *Stettler and Axel, 2009*; *Yaksi et al., 2009*) and upregulate *cfos* during olfactory learning (*Meissner-Bernard et al., 2019*). We then rewired assembly neurons: additional connections were created between assembly neurons, and a matching number of existing connections between non-assembly and assembly neurons were eliminated. The number of input connections per neuron therefore remained unchanged. A new connection between

**Table 5.** Parameters of structured networks.

| | **Figures 2–6** | **Figure 4—figure supplement 3 (mean)** |
|---|---|---|
| Scaled I | $\alpha = 5$, $\chi = 1.4$ | Scaled[adjust]: $\alpha = 3.7$, $\chi = 1.07$ |
| Tuned I | $\alpha = 5$, $\beta = 18$ | / |
| Tuned E+I | $\alpha = 5$, $\beta = 4$, $\gamma = 3$ | Tuned[adjust]: $\alpha = 5$, $\beta = 11.5$, $\gamma = 3$ |

two assembly neurons doubled the synaptic strength $w_{EE}$ if it added to an existing connection. As a result of this rewiring, the connection probability within the assembly increased by a factor $\alpha$ relative to the baseline connection probability.

In *Scaled I* networks, $w_{IE}$ was increased globally by a constant factor $\chi$. In *Tuned* networks, connections were modified between the 100 E neurons of an assembly and the 25 I neurons that were most densely connected to these E neurons, using the same procedure as for E-to-E connections. In *Tuned I* networks, only I-to-E connections were rewired, while in *Tuned E+I networks*, both I-to-E and E-to-I connections were rewired (*Table 5*). Whenever possible, parameters $\alpha$, $\beta$, $\gamma$, and $\chi$ that generated networks with less than 15% change in population firing rates compared to the corresponding *rand* network were selected. In *Figure 6*, two additional assemblies were created in *Scaled I* or *Tuned* networks without adjusting any parameters.

## Observables

All variables were measured in the E population and time-averaged over the first 1.5 s of odor presentations, unless otherwise stated.

1. The *firing rate* is the number of spikes in a time interval $T$ divided by $T$.
2. $g_{OE}$ is the mean conductance change in E neurons triggered by spikes from the OB.
3. $g_{syn}$ is the total synaptic conductance change due to odor stimulation, calculated as the sum of $g_{OE}$, $g_{EE}$, and $g_{IE}$. $g_{EE}$ and $g_{IE}$ are the conductance changes contributed by E and I synapses, respectively.
4. The *percentage of recurrent input* quantifies the average contribution of the recurrent excitatory input to the total excitatory input in E neurons. It was defined for each excitatory neuron as the ratio of the time-averaged $g_{EE}$ to the time-averaged total excitatory conductance ($g_{EE} + g_{OE}$) multiplied by 100. In (2,3,4), the time-averaged E and I synaptic conductances during the 500 ms before odor presentation were subtracted from the E and I conductances measured during odor presentation for each neuron.
5. In addition, we required the Pearson correlation between activity patterns to be close to zero in response to uncorrelated inputs. The Pearson correlation between pairs of activity vectors composed of the firing rate of E neurons was averaged over all possible odor pairs.

## Co-tuning

*Co-tuning* was quantified by two different procedures: (1) For each neuron, we calculated the Pearson correlation between the time-averaged E and I conductances in response to 10 learned odors. (2) As described in *Rupprecht and Friedrich, 2018*, we projected observed pairs of E and I conductances onto a 'balanced' and 'counter-balanced' axis. The balanced axis was obtained by fitting a linear model without constant to the E and I conductances of 4000×10 neuron-learned odor pairs. The resulting model was a constant I/E ratio (~1.2) that defined a membrane potential close to spike threshold. The counter-balanced axis was orthogonal to the balanced axis. For each neuron, synaptic conductances were projected onto these axes and their dispersions quantified by the standard deviations.

## Characterization of population activity in state space

*Principal component analysis* (PCA) was applied to the OB activity patterns of the odor subspace or to the corresponding activity patterns across E neurons in pDp$_{sim}$ (8 × 121 = 968 patterns, each averaged over 1 s).

The *participation ratio* (PR) provides an estimate of the maximal number of principal components (PCs) required to recapitulate the observed neuronal activity. It is defined as $PR = \frac{(\sum_i \lambda_i)^2}{(\sum_i \lambda_i^2)}$, where $\lambda_i$ are the eigenvalues obtained from PCA (variance along each PC).

For angular analyses, we projected activity patterns onto the first 400 PCs, which was the number of PCs required to explain at least 75% of the variance in all networks. We measured the *angle* $\theta$ between the edges connecting the pattern $\mathbf{p}$ evoked by a pure odor to two patterns $\mathbf{s_y}$ and $\mathbf{s_z}$. $\mathbf{s_y}$ and $\mathbf{s_z}$ were patterns evoked by 2 out of the 7 odors that were most similar to the pure odor (168 angles in total). $\theta$ was defined as $\theta_{yz} = \cos^{-1}\left(\frac{(\mathbf{s_z}-\mathbf{p})\cdot(\mathbf{s_y}-\mathbf{p})}{\|\mathbf{s_z}-\mathbf{p}\|\|\mathbf{s_y}-\mathbf{p}\|}\right)$. This metric is sensitive to non-uniform expansion and other non-linear transformations.

The *Mahalanobis distance* $d_{\mathrm{M}}$ is defined as $d_{\mathrm{M}}\left(v, Q\right) = \sqrt{\left(v - \mu\right)^{T} S^{-1} \left(v - \mu\right)}$, where $v$ is a vector representing an activity pattern, and $Q$ a reference class consisting of a distribution of activity patterns with mean $\mu$ and covariance matrix $S$.

## Classification: Four-way forced choice task

To assess the assignment of odor-evoked patterns to representations of pure odors in the odor subspace, we used *quadratic discriminant analysis* (QDA), a non-linear classifier that takes into account the separation and covariance patterns of different classes (*Ghojogh and Crowley, 2019*). The training set consisted of the population response to multiple trials of each of the four pure odors, averaged over the first and second half of the 1 s odor presentation. To increase the number of training data in each of the four odor classes, the training set also included population response to odors that were closely related to the pure odor (Pearson correlation between OB response patterns >0.6). Analyses were performed using subsets of 80 neurons (similar results were obtained using 50–100 neurons). These neurons were randomly selected (50 iterations) either from the (pseudo-) assemblies representing the pure odors (400 E neurons; pseudo-assemblies in *rand* networks and for novel odors) or from the entire population of E neurons. We verified that the data came from a Gaussian mixture model. The trained classifier was then applied to activity patterns evoked by the remaining odors of the subspace (test set; correlation with pure odors <0.6). Each pattern $x$ from the test set was assigned to the class $k$ that maximized the discriminant function.

$\delta_k\left(x\right) = -\frac{1}{2}\log\left|S_k\right| - \frac{1}{2}\left(x - \mu_k\right)^{T} S_k^{-1}\left(x - \mu_k\right) + \log\pi_k$, where $S_k$ is the neuronal covariance matrix of each odor class $k$ and $\pi_k$ is the prior probability of class $k$. This discriminant function is closely related to $d_{\mathrm{M}}$. The subsampling of neurons ensured the invertibility of $S_k$ in *Tuned* and *rand* networks. In *Scaled* networks, $S_k$ is low-rank and therefore not invertible, and QDA cannot be applied.

Template matching (*Figure 5—figure supplement 2*) was performed in a similar way except that each test pattern $x$ was assigned to the class $k$ that maximized the correlation between pattern $x$ and a randomly selected pattern in each class (10 iterations).

## Classification: Odor concentration

As mentioned above, an odor is represented by 150 activated mitral cells, a subset of which remain activated in mixtures. We define the percentage of remaining activated cells as the concentration of the odor. We performed multiple linear regression to predict the concentration of a given odor based on the firing rate of 80 E neurons averaged over the first second of odor presentation. As for QDA, these 80 neurons were randomly selected (50 iterations) from the (pseudo-) assemblies representing the pure odors (400 E neurons; pseudo-assemblies in *rand* networks and for novel odors). We included four or eight trials to equilibrate the number of population responses across the different concentrations ($n$ = 352).

Using the Matlab function *fitcecoc*, we also trained a linear support vector machine model to assess the accuracy of different networks in predicting the concentration of an odor in a mixture. Mixtures were divided into six different concentration classes, with 15% increments (see *Figure 5D*). We included two to six trials to equilibrate the number of population responses within the different classes. The dataset consisted of 162 population responses of 80 randomly selected neurons (20 iterations) and was randomly partitioned in training and testing set (twofold cross-validation, 20 iterations).

## Simulations

Simulations were performed using Matlab and Python. Differential equations were solved using the forward Euler method and an integration time step of $dt$ = 0.1 ms.

## Acknowledgements

We thank the Friedrich lab for insightful discussions. This work was supported by the Novartis Research Foundation, by the European Research Council (ERC) under the European Union's Horizon 2020 research and innovation program (grant agreement no. 742576), and by the Swiss National Science Foundation (grants no. 31003A_172925/1, PCEFP3_202981).

# Additional information

## Funding

| Funder | Grant reference number | Author |
|---|---|---|
| Schweizerischer Nationalfonds zur Förderung der Wissenschaftlichen Forschung | 31003A_172925/1 | Rainer W Friedrich |
| Schweizerischer Nationalfonds zur Förderung der Wissenschaftlichen Forschung | PCEFP3_202981 | Friedemann Zenke |
| European Research Council | 742576 | Rainer W Friedrich |

The funders had no role in study design, data collection and interpretation, or the decision to submit the work for publication.

## Author contributions

Claire Meissner-Bernard, Conceptualization, Software, Formal analysis, Validation, Visualization, Methodology, Writing – original draft, Writing – review and editing, Investigation; Friedemann Zenke, Conceptualization, Resources, Supervision, Methodology, Writing – original draft; Rainer W Friedrich, Conceptualization, Resources, Supervision, Funding acquisition, Investigation, Visualization, Writing – original draft, Project administration, Writing – review and editing

## Author ORCIDs

Claire Meissner-Bernard ⓘ http://orcid.org/0009-0007-2038-8398
Friedemann Zenke ⓘ https://orcid.org/0000-0003-1883-644X
Rainer W Friedrich ⓘ https://orcid.org/0000-0001-9107-0482

Reviewer #1 (Public review): https://doi.org/10.7554/eLife.96303.3.sa1
Reviewer #2 (Public review): https://doi.org/10.7554/eLife.96303.3.sa2
Reviewer #3 (Public review): https://doi.org/10.7554/eLife.96303.3.sa3
Author response https://doi.org/10.7554/eLife.96303.3.sa4

# Additional files

## Supplementary files

MDAR checklist

## Data availability

The current manuscript is a computational study, so no data have been generated for this manuscript. Modeling code is available at https://github.com/clairemb90/pDp-model (copy archived at *Meissner-Bernard, 2024*).

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
