## [Editor Report · eLife Assessment]

This **important** study introduces a biologically constrained model of telencephalic area of adult zebrafish to highlight the significance of precisely balanced memory networks in olfactory processing. The authors provide **compelling** evidence that their model performs better in multiple situations (for e.g. in terms of network stability and shaping the geometry of representations), compared to traditional attractor networks and persistent activity. The work supports recent studies reporting functional E/I subnetworks in several sensory cortexes, and will be of interest to both theoretical and experimental neuroscientists studying network dynamics based on structured excitatory and inhibitory interactions.

---

## [Referee Report · Reviewer #1 (Public review)]

Summary:

Meissner-Bernard et al present a biologically constrained model of telencephalic area of adult zebrafish, a homologous area to the piriform cortex, and argue for the role of precisely balanced memory networks in olfactory processing.

This is interesting as it can add to recent evidence on the presence of functional subnetworks in multiple sensory cortices. It is also important in deviating from traditional accounts of memory systems as attractor networks. Evidence for attractor networks has been found in some systems, like in the head direction circuits in the flies. However, the presence of attractor dynamics in other modalities, like sensory systems, and their role in computation has been more contentious. This work contributes to this active line of research in experimental and computational neuroscience by suggesting that, rather than being represented in attractor networks and persistent activity, olfactory memories might be coded by balanced excitation-inhibitory subnetworks.

Strengths:

The main strength of the work is in: (1) direct link to biological parameters and measurements, (2) good controls and quantification of the results, and (3) comparison across multiple models.

(1) The authors have done a good job of gathering the current experimental information to inform a biological-constrained spiking model of the telencephalic area of adult zebrafish. The results are compared to previous experimental measurements to choose the right regimes of operation.

(2) Multiple quantification metrics and controls are used to support the main conclusions, and to ensure that the key parameters are controlled for - e.g. when comparing across multiple models.

(3) Four specific models (random, scaled I / attractor, and two variant of specific E-I networks - tuned I and tuned E+I) are compared with different metrics, helping to pinpoint which features emerge in which model.

In the revised manuscript, the authors have also:

(a) made a good effort to provide a mechanistic explanation of their results (especially on the mechanism underlying medium amplification in specific E/I network models);

(b) performed a systematic analysis of the parameter space by changing different parameters of E and I neurons (specifically showing that different time constants of E and I neurons do not change the results and therefore the main effects result from connectivity);

(c) added further analysis and discussion on the potential functional and computational significance of balanced specific E-I subnetworks.

These additions substantially strengthen the study, presenting compelling evidence for how networks with specific E-I structure can underpin olfactory processing and memory representations. The findings have potential implications that extend beyond the olfactory system and may be applicable to other neural systems and species.

---

## [Referee Report · Reviewer #2 (Public review)]

Summary:

The authors conducted a comparative analysis of four networks, varying in the presence of excitatory assemblies and the architecture of inhibitory cell assembly connectivity. They found that co-tuned E-I assemblies provide network stability and a continuous representation of input patterns (on locally constrained manifolds), contrasting with networks with global inhibition that result in attractor networks.

Strengths:

The findings presented in this paper are very interesting and cutting-edge. The manuscript effectively conveys the message and presents a creative way to represent high-dimensional inputs and network responses. Particularly, the result regarding the projection of input patterns onto local manifolds and continuous representation of input/memory is very Intriguing and novel. Both computational and experimental neuroscientists would find value in reading the paper.

Weaknesses:

Intuitively, classification (decodability) in discrete attractor networks is much better than in networks with continuous representations. This could also be shown in Figure 5B, along with the performance of the random and tuned E-I networks. The latter networks have the advantage of providing network stability compared to the Scaled I network, but at the cost of reduced network salience and, therefore, reduced input decodability. Thus, tuned E-I networks cannot always perform better than any other network.

---

## [Referee Report · Reviewer #3 (Public review)]

Summary:

This work investigates computational consequences of assemblies containing both excitatory and inhibitory neurons (E/I assembly) in a model with parameters constrained by experimental data from the telencephalic area Dp of zebrafish. The authors show how this precise E/I balance shapes the geometry of neuronal dynamics in comparison to unstructured networks and networks with more global inhibitory balance. Specifically, E/I assemblies lead to the activity being locally restricted onto manifolds - a dynamical structure in-between high-dimensional representations in unstructured networks and discrete attractors in networks with global inhibitory balance. Furthermore, E/I assemblies lead to smoother representations of mixtures of stimuli while those stimuli can still be reliably classified, and allows for more robust learning of additional stimuli.

Strengths:

Since experimental studies do suggest that E/I balance is very precise and E/I assemblies exist, it is important to study the consequences of those connectivity structures on network dynamics. The authors convincingly show that E/I assemblies lead to different geometries of stimulus representation compared to unstructured networks and networks with global inhibition. This finding might open the door for future studies for exploring the functional advantage of these locally defined manifolds, and how other network properties allow to shape those manifolds.

The authors also make sure that their spiking model is well-constrained by experimental data from the zebrafish pDp. Both, spontaneous and odor stimulus triggered spiking activity is within the range of experimental measurements. But the model is also general enough to be potentially applied to findings in other animal models and brain regions.

Weaknesses:

All my previous points have been addressed.

---

## [Author Response]

The following is the authors’ response to the original reviews.

The revised manuscript contains new results and additional text. Major revisions:

(1) Additional simulations and analyses of networks with different biophysical parameters and with identical time constants for E and I neurons (Methods, Supplementary Fig. 5).

(2) Additional simulations and analyses of networks with modifications of connectivity parameters to further analyze effects of E/I assemblies on manifold geometry (Supplementary Fig. 6).

(3) Analysis of synaptic current components (Figure 3 D-F; to analyze mechanism of modest amplification in *Tuned* networks).

(4) More detailed explanation of pattern completion analysis (Results).

(5) Analysis of classification performance of *Scaled* networks (Supplementary Fig.8).

(6) Additional analysis (Figure 5D-F) and discussion (particularly section 'Computational functions of networks with E/I assemblies') of functional benefits of continuous representations in networks with E-I assemblies.

**Public Reviews:**

**Reviewer #1 (Public Review):**
Summary:Meissner-Bernard et al present a biologically constrained model of telencephalic area of adult zebrafish, a homologous area to the piriform cortex, and argue for the role of precisely balanced memory networks in olfactory processing.This is interesting as it can add to recent evidence on the presence of functional subnetworks in multiple sensory cortices. It is also important in deviating from traditional accounts of memory systems as attractor networks. Evidence for attractor networks has been found in some systems, like in the head direction circuits in the flies. However, the presence of attractor dynamics in other modalities, like sensory systems, and their role in computation has been more contentious. This work contributes to this active line of research in experimental and computational neuroscience by suggesting that, rather than being represented in attractor networks and persistent activity, olfactory memories might be coded by balanced excitation-inhibitory subnetworks.Strengths:The main strength of the work is in: (1) direct link to biological parameters and measurements, (2) good controls and quantification of the results, and (3) comparison across multiple models.(1) The authors have done a good job of gathering the current experimental information to inform a biological-constrained spiking model of the telencephalic area of adult zebrafish. The results are compared to previous experimental measurements to choose the right regimes of operation.(2) Multiple quantification metrics and controls are used to support the main conclusions and to ensure that the key parameters are controlled for - e.g. when comparing across multiple models. (3) Four specific models (random, scaled I / attractor, and two variant of specific E-I networks - tuned I and tuned E+I) are compared with different metrics, helping to pinpoint which features emerge in which model.Weaknesses:Major problems with the work are: (1) mechanistic explanation of the results in specific E-I networks, (2) parameter exploration, and (3) the functional significance of the specific E-I model.(1) The main problem with the paper is a lack of mechanistic analysis of the models. The models are treated like biological entities and only tested with different assays and metrics to describe their different features (e.g. different geometry of representation in Fig. 4). Given that all the key parameters of the models are known and can be changed (unlike biological networks), it is expected to provide a more analytical account of why specific networks show the reported results. For instance, what is the key mechanism for medium amplification in specific E/I network models (Fig. 3)? How does the specific geometry of representation/manifolds (in Fig. 4) emerge in terms of excitatory-inhibitory interactions, and what are the main mechanisms/parameters? Mechanistic account and analysis of these results are missing in the current version of the paper.

We agree that further mechanistic insights would be of interest and addressed this issue at different levels:

(1) Biophysical parameters: to determine whether network behavior depends on specific choices of biophysical parameters in E and I neurons we equalized biophysical parameters across neuron types. The main observations are unchanged, suggesting that the observed effects depend primarily on network connectivity (see also response to comment [2]).

(2) Mechanism of modest amplification in E/I assemblies: analyzing the different components of the synaptic currents demonstrate that the modest amplification of activity in *Tuned* networks results from an 'imperfect' balance of *recurrent* excitation and inhibition within assemblies (see new Figures 3D-F and text p.7). Hence, E/I co-tuning substantially reduces the net amplification in *Tuned* networks as compared to *Scaled* networks, thus preventing discrete attractor dynamics and stabilizing network activity, but a modest amplification still occurs, consistent with biological observations.

(3) Representational geometry: to obtain insights into the network mechanisms underlying effects of E/I assemblies on the geometry of population activity we tested the hypothesis that geometrical changes depend, at least in part, on the modest amplification of activity within E/I assemblies (see Supplementary Figure 6). We changed model parameters to either prevent the modest amplification in *Tuned* networks (increasing I-to-E connectivity within assemblies) or introduce a modest amplification in subsets of neurons by other mechanisms (concentration-dependent increase in the excitability of pseudo-assembly neurons; *Scaled* I networks with reduced connectivity within assemblies). Manipulations that introduced a modest, input-dependent amplification in neuronal subsets had geometrical effects similar to those observed in *Tuned* networks, whereas manipulations that prevented a modest amplification abolished these effects (Supplementary Figure 6). Note however that these manipulations generated different firing rate distributions. These results provide a starting point for more detailed analyses of the relationship between network connectivity and representational geometry (see p.12).

In summary, our additional analyses indicate that effects of E/I assemblies on representational geometry depend primarily on network connectivity, rather than specific biophysical parameters, and that the resulting modest amplification of activity within assemblies makes an important contribution. Further analyses may reveal more specific relationships between E/I assemblies and representational geometry, but such analyses are beyond the scope of this study.

(2) The second major issue with the study is a lack of systematic exploration and analysis of the parameter space. Some parameters are biologically constrained, but not all the parameters. For instance, it is not clear what the justification for the choice of synaptic time scales are (with E synaptic time constants being larger than inhibition: tau_syn_i = 10 ms, tau_syn_E = 30 ms). How would the results change if they are varying these - and other unconstrained - parameters? It is important to show how the main results, especially the manifold localisation, would change by doing a systematic exploration of the key parameters and performing some sensitivity analysis. This would also help to see how robust the results are, which parameters are more important and which parameters are less relevant, and to shed light on the key mechanisms.

We thank the reviewer for raising this point. We chose a relatively slow time constant for excitatory synapses because experimental data indicate that excitatory synaptic currents in Dp and piriform cortex contain a prominent NMDA component. Nevertheless, to assess whether network behavior depends on specific choices of biophysical parameters in E and I neurons, we have performed additional simulations with equal synaptic time constants and equal biophysical parameters for all neurons. Each neuron also received the same number of inputs from each population (see revised Methods). Results were similar to those observed previously (Supplementary Fig.5 and p.9 of main text). We therefore conclude that the main effects observed in *Tuned* networks cannot be explained by differences in biophysical parameters between E and I neurons but is primarily a consequence of network connectivity.

(3) It is not clear what the main functional advantage of the specific E-I network model is compared to random networks. In terms of activity, they show that specific E-I networks amplify the input more than random networks (Fig. 3). But when it comes to classification, the effect seems to be very small (Fig. 5c). Description of different geometry of representation and manifold localization in specific networks compared to random networks is good, but it is more of an illustration of different activity patterns than proving a functional benefit for the network. The reader is still left with the question of what major functional benefits (in terms of computational/biological processing) should be expected from these networks, if they are to be a good model for olfactory processing and learning.One possibility for instance might be that the tasks used here are too easy to reveal the main benefits of the specific models - and more complex tasks would be needed to assess the functional enhancement (e.g. more noisy conditions or more combination of odours). It would be good to show this more clearly - or at least discuss it in relation to computation and function.

In the previous manuscript, the analysis of potential computational benefits other than pattern classification was limited and the discussion of this issue was condensed into a single itemized paragraph to avoid excessive speculation. Although a thorough analysis of potential computational benefits exceeds the scope of a single paper, we agree with the reviewer that this issue is of interest and therefore added additional analyses and discussion.

In the initial manuscript we analyzed pattern classification primarily to investigate whether *Tuned* networks can support this function at all, given that they do not exhibit discrete attractor states. We found this to be the case, which we consider a first important result.

Furthermore, we found that precise balance of E/I assemblies can protect networks against catastrophic firing rate instabilities when assemblies are added sequentially, as in continual learning. Results from these simulations are now described and discussed in more detail (see Results p.11 and Discussion p.13).

In the revised manuscript, we now also examine additional potential benefits of *Tuned* networks and discuss them in more detail (see new Figure 5D-F and text p.11). One hypothesis is that continuous representations provide a distance metric between a given input and relevant (learned) stimuli. To address this hypothesis, we (1) performed regression analysis and (2) trained support vector machines (SVMs) to predict the concentration of a given odor in a mixture based on population activity. In both cases, *Tuned E+I* networks outperformed *Scaled* and _rand n_etworks in predicting the concentration of learned odors across a wide range mixtures (Figure 5D-F). E/I assemblies therefore support the quantification of learned odors within mixtures or, more generally, assessments of how strongly a (potentially complex) input is related to relevant odors stored in memory. Such a metric assessment of stimulus quality is not well supported by discrete attractor networks because inputs are mapped onto discrete network states.

The observation that *Tuned* networks do not map inputs onto discrete outputs indicates that such networks do not classify inputs as distinct items. Nonetheless, the observed geometrical modifications of continuous representations support the classification of learned inputs or the assessment of metric relationships by hypothetical readout neurons. Geometrical modifications of odor representations may therefore serve as one of multiple steps in multi-layer computations for pattern classification (and/or other computations). In this scenario, the transformation of odor representations in Dp may be seen as related to transformations of representations between different layers in artificial networks, which collectively perform a given task (notwithstanding obvious structural and mechanistic differences between artificial and biological networks). In other words, geometrical transformations of representations in *Tuned* networks may overrepresent learned (relevant) information at the expense of other information and thereby support further learning processes in other brain areas. An obvious corollary of this scenario is that Dp does not perform odor classification *per se* based on inputs from the olfactory bulb but reformats representations of odor space based on experience to support computational tasks as part of a larger system. This scenario is now explicitly discussed (p.14).

**Reviewer #2 (Public Review):**
Summary:The authors conducted a comparative analysis of four networks, varying in the presence of excitatory assemblies and the architecture of inhibitory cell assembly connectivity. They found that co-tuned E-I assemblies provide network stability and a continuous representation of input patterns (on locally constrained manifolds), contrasting with networks with global inhibition that result in attractor networks.Strengths:The findings presented in this paper are very interesting and cutting-edge. The manuscript effectively conveys the message and presents a creative way to represent high-dimensional inputs and network responses. Particularly, the result regarding the projection of input patterns onto local manifolds and continuous representation of input/memory is very Intriguing and novel. Both computational and experimental neuroscientists would find value in reading the paper.Weaknesses:that have continuous representations. This could also be shown in Figure 5B, along with the performance of the random and tuned E-I networks. The latter networks have the advantage of providing network stability compared to the Scaled I network, but at the cost of reduced network salience and, therefore, reduced input decodability. The authors may consider designing a decoder to quantify and compare the classification performance of all four networks.

We have now quantified classification by networks with discrete attractor dynamics (*Scaled*) along with other networks. However, because the neuronal covariance matrix for such networks is low rank and not invertible, pattern classification cannot be analyzed by QDA as in Figure 5B. We therefore classified patterns from the odor subspace by template matching, assigning test patterns to one of the four classes based on correlations (see Supplementary Figure 8). As expected, *Scaled* networks performed well, but they did not outperform *Tuned* networks. Moreover, the performance of *Scaled* networks, but not *Tuned* networks, depended on the order in which odors were presented to the network. This hysteresis effect is a direct consequence of persistent attractor states and decreased the general classification performance of *Scaled* networks (see Supplementary Figure 8 for details). These results confirm the prediction that networks with discrete attractor states can efficiently classify inputs, but also reveal disadvantages arising from attractor dynamics. Moreover, the results indicate that the classification performance of *Tuned* networks is also high under the given task conditions, which simulate a biologically realistic scenario.

We would also like to emphasize that classification may not be the only task, and perhaps not even a main task, of Dp/piriform cortex or other memory networks with E/I assemblies. Conceivably, other computations could include metric assessments of inputs relative to learned inputs or additional learning-related computations. Please see our response to comment (3) of reviewer 1 for a further discussion of this issue.

Networks featuring E/I assemblies could potentially represent multistable attractors by exploring the parameter space for their reciprocal connectivity and connectivity with the rest of the network. However, for co-tuned E-I networks, the scope for achieving multistability is relatively constrained compared to networks employing global or lateral inhibition between assemblies. It would be good if the authors mentioned this in the discussion. Also, the fact that reciprocal inhibition increases network stability has been shown before and should be cited in the statements addressing network stability (e.g., some of the citations in the manuscript, including Rost et al. 2018, Lagzi & Fairhall 2022, and Vogels et al. 2011 have shown this).

We thank the reviewer for this comment. We now explicitly discuss multistability (see p. 12) and refer to additional references in the statements addressing network stability.

Providing raster plots of the pDp network for familiar and novel inputs would help with understanding the claims regarding continuous versus discrete representation of inputs, allowing readers to visualize the activity patterns of the four different networks. (similar to Figure 1B).

We thank the reviewer for this suggestion. We have added raster plots of responses to both familiar and novel inputs in the revised manuscript (Figure 2D and Supplementary Figure 4A).

**Reviewer #3 (Public Review):**
Summary:This work investigates the computational consequences of assemblies containing both excitatory and inhibitory neurons (E/I assembly) in a model with parameters constrained by experimental data from the telencephalic area Dp of zebrafish. The authors show how this precise E/I balance shapes the geometry of neuronal dynamics in comparison to unstructured networks and networks with more global inhibitory balance. Specifically, E/I assemblies lead to the activity being locally restricted onto manifolds - a dynamical structure in between high-dimensional representations in unstructured networks and discrete attractors in networks with global inhibitory balance. Furthermore, E/I assemblies lead to smoother representations of mixtures of stimuli while those stimuli can still be reliably classified, and allow for more robust learning of additional stimuli.Strengths:Since experimental studies do suggest that E/I balance is very precise and E/I assemblies exist, it is important to study the consequences of those connectivity structures on network dynamics. The authors convincingly show that E/I assemblies lead to different geometries of stimulus representation compared to unstructured networks and networks with global inhibition. This finding might open the door for future studies for exploring the functional advantage of these locally defined manifolds, and how other network properties allow to shape those manifolds.The authors also make sure that their spiking model is well-constrained by experimental data from the zebrafish pDp. Both spontaneous and odor stimulus triggered spiking activity is within the range of experimental measurements. But the model is also general enough to be potentially applied to findings in other animal models and brain regions.Weaknesses:I find the point about pattern completion a bit confusing. In Fig. 3 the authors argue that only the Scaled I network can lead to pattern completion for morphed inputs since the output correlations are higher than the input correlations. For me, this sounds less like the network can perform pattern completion but it can nonlinearly increase the output correlations. Furthermore, in Suppl. Fig. 3 the authors show that activating half the assembly does lead to pattern completion in the sense that also non-activated assembly cells become highly active and that this pattern completion can be seen for Scaled I, Tuned E+I, and Tuned I networks. These two results seem a bit contradictory to me and require further clarification, and the authors might want to clarify how exactly they define pattern completion.

We believe that this comment concerns a semantic misunderstanding and apologize for any lack of clarity. We added a definition of pattern completion in the text: '…the retrieval of the whole memory from noisy or corrupted versions of the learned input.'. Pattern completion may be assessed using different procedures. In computational studies, it is often analyzed by delivering input to a subset of the assembly neurons which store a given memory (partial activation). Under these conditions, we find recruitment of the entire assembly in all structured networks, as demonstrated in Supplementary Figure 3. However, these conditions are unlikely to occur during odor presentation because the majority of neurons do not receive any input.

Another more biologically motivated approach to assess pattern completion is to gradually modify a realistic odor input into a learned input, thereby gradually increasing the overlap between the two inputs. This approach had been used previously in experimental studies (references added to the text p.6). In the presence of assemblies, recurrent connectivity is expected to recruit assembly neurons (and thus retrieve the stored pattern) more efficiently as the learned pattern is approached. This should result in a nonlinear increase in the similarity between the evoked and the learned activity pattern. This signature was prominent in *Scaled* networks but not in *Tuned* or *rand* networks. Obviously, the underlying procedure is different from the partial activation of the assembly described above because input patterns target many neurons (including neurons outside assemblies) and exhibit a biologically realistic distribution of activity. However, this approach has also been referred to as 'pattern completion' in the neuroscience literature, which may be the source of semantic confusion here. To clarify the difference between these approaches we have now revised the text and explicitly described each procedure in more detail (see p.6).

The authors argue that Tuned E+I networks have several advantages over Scaled I networks. While I agree with the authors that in some cases adding this localized E/I balance is beneficial, I believe that a more rigorous comparison between Tuned E+I networks and Scaled I networks is needed: quantification of variance (Fig. 4G) and angle distributions (Fig. 4H) should also be shown for the Scaled I network. Similarly in Fig. 5, what is the Mahalanobis distance for Scaled I networks and how well can the Scaled I network be classified compared to the Tuned E+I network? I suspect that the Scaled I network will actually be better at classifying odors compared to the E+I network. The authors might want to speculate about the benefit of having networks with both sources of inhibition (local and global) and hence being able to switch between locally defined manifolds and discrete attractor states.

We agree that a more rigorous comparison of Tuned and Scaled networks would be of interest. We have added the variance analysis (Fig 4G) and angle distributions (Fig. 4H) for both Tuned I and Scaled networks. However, the Mahalanobis distances and Quadratic Discriminant Analysis cannot be applied to Scaled networks because their neuronal covariance matrix is low rank and not invertible_._ To nevertheless compare these networks, we performed template matching by assigning test patterns to one of the four odor classes based on correlations to template patterns (Supplementary Figure 8; see also response to the first comment of reviewer 2). Interestingly, *Scaled* networks performed well at classification but did not outperform *Tuned* networks, and exhibited disadvantages arising from attractor dynamics (Supplementary Figure 8; see also response to the first comment of reviewer 2)*.* Furthermore, in further analyses we found that continuous representational manifolds support metric assessments of inputs relative to learned odors, which cannot be achieved by discrete representations. These results are now shown in Figure 5D-E and discussed explicitly in the text on p.11 (see also response to comment 3 of reviewer 1).

We preferred not to add a sentence in the Discussion about benefits of networks having both sources of inhibition, as we find this a bit too speculative.

At a few points in the manuscript, the authors use statements without actually providing evidence in terms of a Figure. Often the authors themselves acknowledge this, by adding the term "not shown" to the end of the sentence. I believe it will be helpful to the reader to be provided with figures or panels in support of the statements.

Thank you for this comment. We have provided additional data figures to support the following statements:

'd_M_ was again increased upon learning, particularly between learned odors and reference classes representing other odors (Supplementary Figure 9)'

'decreasing amplification in assemblies of Scaled networks changed transformations towards the intermediate behavior, albeit with broader firing rate distributions than in Tuned networks (Supplementary Figure 6 B)'

**Recommendations for the authors:**

**Reviewer #1 (Recommendations For The Authors):**
Meissner-Bernard et al present a biologically constrained model of telencephalic area of adult zebrafish, a homologous area to the piriform cortex, and argue for the role of precisely balanced memory networks in olfactory processing.This is interesting as it can add to recent evidence on the presence of functional subnetworks in multiple sensory cortices. It is also important in deviating from traditional accounts of memory systems as attractor networks. Evidence for attractor networks has been found in some systems, like in the head direction circuits in the flies. However, the presence of attractor dynamics in other modalities, like sensory systems, and their role in computation has been more contentious. This work contributes to this active line of research in experimental and computational neuroscience by suggesting that, rather than being represented in attractor networks and persistent activity, olfactory memories might be coded by balanced excitation-inhibitory subnetworks.The paper is generally well-written, the figures are informative and of good quality, and multiple approaches and metrics have been used to test and support the main results of the paper.The main strength of the work is in: (1) direct link to biological parameters and measurements, (2) good controls and quantification of the results, and (3) comparison across multiple models.(1) The authors have done a good job of gathering the current experimental information to inform a biological-constrained spiking model of the telencephalic area of adult zebrafish. The results are compared to previous experimental measurements to choose the right regimes of operation.(2) Multiple quantification metrics and controls are used to support the main conclusions and to ensure that the key parameters are controlled for - e.g. when comparing across multiple models. (3) Four specific models (random, scaled I / attractor, and two variant of specific E-I networks - tuned I and tuned E+I) are compared with different metrics, helping to pinpoint which features emerge in which model.Major problems with the work are: (1) mechanistic explanation of the results in specific E-I networks, (2) parameter exploration, and (3) the functional significance of the specific E-I model.(1) The main problem with the paper is a lack of mechanistic analysis of the models. The models are treated like biological entities and only tested with different assays and metrics to describe their different features (e.g. different geometry of representation in Fig. 4). Given that all the key parameters of the models are known and can be changed (unlike biological networks), it is expected to provide a more analytical account of why specific networks show the reported results. For instance, what is the key mechanism for medium amplification in specific E/I network models (Fig. 3)? How does the specific geometry of representation/manifolds (in Fig. 4) emerge in terms of excitatory-inhibitory interactions, and what are the main mechanisms/parameters? Mechanistic account and analysis of these results are missing in the current version of the paper.Precise balancing of excitation and inhibition in subnetworks would lead to the cancellation of specific dynamical modes responsible for the amplification of responses (hence, deviating from the attractor dynamics with an unstable specific mode). What is the key difference in the specific E/I networks here (tuned I or/and tuned E+I) which make them stand between random and attractor networks? Excitatory and inhibitory neurons have different parameters in the model (Table 1). Time constants of inhibitory and excitatory synapses are also different (P. 13). Are these parameters causing networks to be effectively more excitation dominated (hence deviating from a random spectrum which would be expected from a precisely balanced E/I network, with exactly the same parameters of E and I neurons)? It is necessary to analyse the network models, describe the key mechanism for their amplification, and pinpoint the key differences between E and I neurons which are crucial for this.

To address these comments we performed additional simulations and analyses at different levels. Please see our reply to comment (1) of the public review (reviewer 1) for a detailed description. We thank the reviewer for these constructive comments.

(2) The second major issue with the study is a lack of systematic exploration and analysis of the parameter space. Some parameters are biologically constrained, but not all the parameters. For instance, it is not clear what the justification for the choice of synaptic time scales are (with E synaptic time constants being larger than inhibition: tau_syn_i = 10 ms, tau_syn_E = 30 ms). How would the results change if they are varying these - and other unconstrained - parameters? It is important to show how the main results, especially the manifold localisation, would change by doing a systematic exploration of the key parameters and performing some sensitivity analysis. This would also help to see how robust the results are, which parameters are more important and which parameters are less relevant, and to shed light on the key mechanisms.

We thank the reviewer for this comment. We have now carried out additional simulations with equal time constants for all neurons. Please see our reply to the public review for more details (comment 2 of reviewer 1).

(3) It is not clear what the main functional advantage of the specific E-I network model is compared to random networks. In terms of activity, they show that specific E-I networks amplify the input more than random networks (Fig. 3). But when it comes to classification, the effect seems to be very small (Fig. 5c). Description of different geometry of representation and manifold localization in specific networks compared to random networks is good, but it is more of an illustration of different activity patterns than proving a functional benefit for the network. The reader is still left with the question of what major functional benefits (in terms of computational/biological processing) should be expected from these networks, if they are to be a good model for olfactory processing and learning.

One possibility for instance might be that the tasks used here are too easy to reveal the main benefits of the specific models - and more complex tasks would be needed to assess the functional enhancement (e.g. more noisy conditions or more combination of odours). It would be good to show this more clearly - or at least discuss it in relation to computation and function.

Please see our reply to the public review (comment 3 of reviewer 1).

Specific comments:Abstract: "resulting in continuous representations that reflected both relatedness of inputs and *an individual's experience*"It didn't become apparent from the text or the model where the role of "individual's experience" component (or "internal representations" - in the next line) was introduced or shown (apart from a couple of lines in the Discussion)

We consider the scenario that that assemblies are the outcome of an experience-dependent plasticity process. To clarify this, we have now made a small addition to the text: 'Biological memory networks are thought to store information by experience-dependent changes in the synaptic connectivity between assemblies of neurons.'.

P. 2: "The resulting state of "precise" synaptic balance stabilizes firing rates because inhomogeneities or fluctuations in excitation are tracked by correlated inhibition"It is not clear what the "inhomogeneities" specifically refers to - they can be temporal, or they can refer to the quenched noise of connectivity, for instance. Please clarify what you mean.

The statement has been modified to be more precise: '…'precise' synaptic balance stabilizes firing rates because inhomogeneities in excitation across the population or temporal variations in excitation are tracked by correlated inhibition…'.

P. 3 (and Methods): When odour stimulus is simulated in the OB, the activity of a fraction of mitral cells is increased (10% to 15 Hz) - but also a fraction of mitral cells is suppressed (5% to 2 Hz). What is the biological motivation or reference for this? It is not provided. Is it needed for the results? Also, it is not explained how the suppressed 5% are chosen (e.g. randomly, without any relation to the increased cells?).

We thank the reviewer for this comment. These changes in activity directly reflect experimental observations. We apologize that we forgot to include the references reporting these observations (Friedrich and Laurent, 2001 and 2004); this is now fixed.

In our simulation, OB neurons do not interact with each other, and the suppressed 5% were indeed randomly selected. We changed the text in Methods accordingly to read: 'An additional 75 randomly selected mitral cells were inhibited'

P. 4, L. 1-2: "... sparsely connected integrate-and-fire neurons with conductance-based synapses (connection probability {less than or equal to}5%)."Specify the connection probability of specific subtypes (EE, EI, IE, II).

We now refer to the Methods section, where this information can be found.

'... conductance-based synapses (connection probability ≤5%, Methods)'

P. 4, L. 6-7: "Population activity was odor-specific and activity patterns evoked by uncorrelated OB inputs remained uncorrelated in Dp (Figure 1H)"What would happen to correlated OB inputs (e.g. as a result of mixture of two overlapping odours) in this baseline state of the network (before memories being introduced to it)? It would be good to know this, as it sheds light on the initial operating regime of the network in terms of E/I balance and decorrelation of inputs.

This information was present in the original manuscript at (Figure 3) but we improved the writing to further clarify this issue: ' (…) we morphed a novel odor into a learned odor (Figure 3A), or a learned odor into another learned odor (Supplementary Figure 3B), and quantified the similarity between morphed and learned odors by the Pearson correlation of the OB activity patterns (input correlation). We then compared input correlations to the corresponding pattern correlations among E neurons in Dp (output correlation). In *rand* networks, output correlations increased linearly with input correlations but did not exceed them (Figure 3B and Supplementary Figure 3B)'

P. 4, L. 12-13: "Shuffling spike times of inhibitory neurons resulted in runaway activity with a probability of ~80%, .." Where is this shown?(There are other occasions too in the paper where references to the supporting figures are missing).

We now provide the statistics: 'Shuffling spike times of inhibitory neurons resulted in runaway activity with a probability of 0.79 ± 0.20'

P. 4: "In each network, we created 15 assemblies representing uncorrelated odors. As a consequence, ~30% of E neurons were part of an assembly ..."15 x 100 / 4000 = 37.5% - so it's closer to 40% than 30%. Unless there is some overlap?

Yes: despite odors being uncorrelated and connectivity being random, some neurons (6 % of E neurons) belong to more than one assembly.

P. 4: "When a reached a critical value of ~6, networks became unstable and generated runaway activity (Figure 2B)."Can this transition point be calculated or estimated from the network parameters, and linked to the underlying mechanisms causing it?

We thank the reviewer for this interesting question. The unstability arises when inhibitions fails to counterbalance efficiently the increased recurrent excitation within Dp. The transition point is difficult to estimate, as it can depend on several parameters, including the probability of E to E connections, their strength, assembly size, and others. We have therefore not attempted to estimate it analytically.

P. 4: "Hence, non-specific scaling of inhibition resulted in a divergence of firing rates that exhausted the dynamic range of individual neurons in the population, implying that homeostatic global inhibition is insufficient to maintain a stable firing rate distribution."I don't think this is justified based on the results and figures presented here (Fig. 2E) - the interpretation is a bit strong and biased towards the conclusions the authors want to draw.

To more clearly illustrate the finding that in *Scaled* networks, assembly neurons are highly active (close to maximal realistic firing rates) whereas non-assembly neurons are nearly silent we have now added Supplementary Fig. 2B. Moreover, we have toned down the text: 'Hence, non-specific scaling of inhibition resulted in a large and biologically unrealistic divergence of firing rates (Supplementary Figure 2B) that nearly exhausted the dynamic range of individual neurons in the population, indicating that homeostatic global inhibition is insufficient to maintain a stable firing rate distribution'

P. 5, third paragraph: Description of Figure 2I, inset is needed, either in the text or caption.

The inset is now referred to in the text: 'we projected synaptic conductances of each neuron onto a line representing the E/I ratio expected in a balanced network ('balanced axis') and onto an orthogonal line ('counter-balanced axis'; Figure 2I inset, Methods).'

P. 5, last paragraph: another example of writing about results without showing/referring to the corresponding figures:"In rand networks, firing rates increased after stimulus onset and rapidly returned to a low baseline after stimulus offset. Correlations between activity patterns evoked by the same odor at different time points and in different trials were positive but substantially lower than unity, indicating high variability ..."And the continuation with similar lack of references on P. 6:"Scaled networks responded to learned odors with persistent firing of assembly neurons and high pattern correlations across trials and time, implying attractor dynamics (Hopfield, 1982; Khona and Fiete, 2022), whereas Tuned networks exhibited transient responses and modest pattern correlations similar to rand networks."Please go through the Results and fix the references to the corresponding figures on all instances.

We thank the reviewer for pointing out these overlooked figure references, which are now fixed.

P. 8: "These observations further support the conclusion that E/I assemblies locally constrain neuronal dynamics onto manifolds."As discussed in the general major points, mechanistic explanation in terms of how the interaction of E/I dynamics leads to this is missing.

As discussed in the reply to the public review (comment 3 of reviewer 1), we have now provided more mechanistic analyses of our observations.

P. 9: "Hence, E/I assemblies enhanced the classification of inputs related to learned patterns." The effect seems to be very small. Also, any explanation for why for low test-target correlation the effect is negative (random doing better than tuned E/I)?

The size of the effect (plearned – pnovel = 0.074; difference of means; Figure 5C) may appear small in terms of absolute probability, but it is substantial relative to the maximum possible increase (1 – p_novel_ = 0.133; Figure 5C). The fact that for low test-target correlations the effect is negative is a direct consequence of the positive effect for high test-target correlations and the presence of 2 learned odors in the 4-way forced choice task.

P. 9: "In Scaled I networks, creating two additional memories resulted in a substantial increase in firing rates, particularly in response to the learned and related odors" Where is this shown? Please refer to the figure.

We thank the reviewer again for pointing this out. We forgot to include a reference to the relevant figure which has now been added in the revised manuscript (Figure 6C).

P. 10: "The resulting Tuned networks reproduced additional experimental observations that were not used as constraints including irregular firing patterns, lower output than input correlations, and the absence of persistent activity"It is difficult to present these as "additional experimental observations", as all of them are negative, and can exist in random networks too - hence cannot be used as biological evidence in favour of specific E/I networks when compared to random networks.

We agree with the reviewer that these additional experimental observations cannot be used as biological evidence favouring Tuned E+I networks over random networks. We here just wanted to point out that additional observations which we did not take into account to fit the model are not invalidating the existence of E-I assemblies in biological networks. As assemblies tend to result in persistent activity in other types of networks, we feel that this observation is worth pointing out.

Methods:P. 13: Describe the parameters of Eq. 2 after the equation.

Done.

P. 13: "The time constants of inhibitory and excitatory synapses were 10 ms and 30 ms, respectively."What is the (biological) justification for the choice of these parameters?How would varying them affect the main results (e.g. local manifolds)?

We chose a relatively slow time constant for excitatory synapses because experimental data indicate that excitatory synaptic currents in Dp and piriform cortex contain a prominent NMDA component. We have now also simulated networks with equal time constants for excitatory and inhibitory synapses and equal biophysical parameters for excitatory and inhibitory neurons, which did not affect the main results (see also reply to the public review: comment 2 of reviewer 1).

P. 14: "Care was also taken to ensure that the variation in the number of output connections was low across neurons" How exactly?

More detailed explanations have now been added in the Methods section: 'connections of a presynaptic neuron *y* to postsynaptic neurons *x* were randomly deleted when their total number exceeded the average number of output connections by ≥5%, or added when they were lower by ≥5%.'

**Reviewer #2 (Recommendations For The Authors):**
Congratulations on the great and interesting work! The results were nicely presented and the idea of continuous encoding on manifolds is very interesting. To improve the quality of the paper, in addition to the major points raised in the public review, here are some more detailed comments for the paper:(1) Generally, citations have to improve. Spiking networks with excitatory assemblies and different architectures of inhibitory populations have been studied before, and the claim about improved network stability in co-tuned E-I networks has been made in the following papers that need to be correctly cited:• Vogels TP, Sprekeler H, Zenke F, Clopath C, Gerstner W. 2011. Inhibitory Plasticity Balances Excitation and Inhibition in Sensory Pathways and Memory Networks. Science 334:1-7. doi:10.1126/science.1212991 (mentions that emerging precise balance on the synaptic weights can result in the overall network stability)• Lagzi F, Bustos MC, Oswald AM, Doiron B. 2021. Assembly formation is stabilized by Parvalbumin neurons and accelerated by Somatostatin neurons. bioRxiv doi: https://doi.org/10.1101/2021.09.06.459211 (among other things, contrasts stability and competition which arises from multistable networks with global inhibition and reciprocal inhibition) • Rost T, Deger M, Nawrot MP. 2018. Winnerless competition in clustered balanced networks: inhibitory assemblies do the trick. Biol Cybern 112:81-98. doi:10.1007/s00422-017-0737-7 (compares different architectures of inhibition and their effects on network dynamics)• Lagzi F, Fairhall A. 2022. Tuned inhibitory firing rate and connection weights as emergent network properties. bioRxiv 2022.04.12.488114. doi:10.1101/2022.04.12.488114 (here, see the eigenvalue and UMAP analysis for a network with global inhibition and E/I assemblies)Additionally, there are lots of pioneering work about tracking of excitatory synaptic inputs by inhibitory populations, that are missing in references. Also, experimental work that show existence of cell assemblies in the brain are largely missing. On the other hand, some references that do not fit the focus of the statements have been incorrectly cited.The authors may consider referencing the following more pertinent studies on spiking networks to support the statement regarding attractor dynamics in the first paragraph in the Introduction (the current citations of Hopfield and Kohonen are for rate-based networks):• Wong, K.-F., & Wang, X.-J. (2006). A recurrent network mechanism of time integration in perceptual decisions. Journal of Neuroscience, 26(4), 1314-1328. https://doi.org/10.1523/JNEUROSCI.3733-05.2006• Wang, X.-J. (2008). Decision making in recurrent neuronal circuits. Neuron, 60(2), 215-234. https://doi.org/10.1016/j.neuron.2008.09.034• F. Lagzi, & S. Rotter. (2015). Dynamics of competition between subnetworks of spiking neuronal networks in the balanced state. PloS One.• Goldman-Rakic, P. S. (1995). Cellular basis of working memory. Neuron, 14(3), 477-485.• Rost T, Deger M, Nawrot MP. 2018. Winnerless competition in clustered balanced networks: inhibitory assemblies do the trick. Biol Cybern 112:81-98. doi:10.1007/s00422-017-0737-7.• Amit DJ, Tsodyks M (1991) Quantitative study of attractor neural network retrieving at low spike rates: I. substrate-spikes, rates and neuronal gain. Network 2:259-273.• Mazzucato, L., Fontanini, A., & La Camera, G. (2015). Dynamics of Multistable States during Ongoing and Evoked Cortical Activity. Journal of Neuroscience, 35(21), 8214-8231.

We thank the reviewer for the references suggestions. We have carefully reviewed the reference list and made the following changes, which we hope address the reviewer’s concerns:

(1) We adjusted References about network stability in co-tuned E-I networks.

(2) We added the Lagzi & Rotter (2015), Amit et al. (1991), Mazzucato et al. (2015) and GoldmanRakic (1995) papers in the Introduction as studies on attractor dynamics in spiking neural networks. We preferred to omit the two X.J Wang papers, as they describe attractors in decision making rather than memory processes.

(3) We added the Ko et al. 2011 paper as experimental evidence for assemblies in the brain. In our view, there are few experimental studies showing the existence of cell assemblies in the brain, which we distinguish from cell ensembles, group of coactive neurons.

(4) We also included Hennequin 2018, Brunel 2000, Lagzi et al. 2021 and Eckmann et al. 2024, which we had not cited in the initial manuscript.

(5) We removed the Wiechert et al. 2010 reference as it does not support the statement about geometry-preserving transformation by random networks.

(2) The gist of the paper is about how the architecture of inhibition (reciprocal vs. global in this case) can determine network stability and salient responses (related to multistable attractors and variations) for classification purposes. It would improve the narrative of the paper if this point is raised in the Introduction and Discussion section. Also see a relevant paper that addresses this point here:Lagzi F, Bustos MC, Oswald AM, Doiron B. 2021. Assembly formation is stabilized by Parvalbumin neurons and accelerated by Somatostatin neurons. bioRxiv doi: https://doi.org/10.1101/2021.09.06.459211

Classification has long been proposed to be a function of piriform cortex and autoassociative memory networks in general, and we consider it important. However, the computational function of Dp or piriform cortex is still poorly understood, and we do not focus only on odor classification as a possibility. In fact, continuous representational manifolds also support other functions such as the quantification of distance relationships of an input to previously memorized stimuli, or multi-layer network computations (including classification). In the revised manuscript, we have performed additional analyses to explore these notions in more detail, as explained above (response to public reviews, comment 3 of reviewer 1). Furthermore, we have now expanded the discussion of potential computational functions of *Tuned* networks and explicitly discuss classification but also other potential functions.

(3) A plot for the values of the inhibitory conductances in Figure 1 would complete the analysis for that section.

In Figure 1, we decided to only show the conductances that we use to fit our model, namely the afferent and total synaptic conductances. As the values of the inhibitory conductances can be derived from panel E, we refrained from plotting them separately for the sake of simplicity.

(4) How did the authors calculate correlations between activity patterns as a function of time in Figure 2E, bottom row? Does the color represent correlation coefficient (which should not be time dependent) or is it a correlation function? This should be explained in the Methods section.

The color represents the Pearson correlation coefficient between activity patterns within a narrow time window (100 ms). We updated the Figure legend to clarify this: 'Mean correlation between activity patterns evoked by a learned odor at different time points during odor presentation. Correlation coefficients were calculated between pairs of activity vectors composed of the mean firing rates of E neurons in 100 ms time bins. Activity vectors were taken from the same or different trials, except for the diagonal, where only patterns from different trials were considered.'

(5) Figure 3 needs more clarification (both in the main text and the figure caption). It is not clear what the axes are exactly, and why the network responses for familiar and novel inputs are different. The gray shaded area in panel B needs more explanation as well.

We thank the reviewer for the comment. We have improved Figure 3A, the figure caption, as well as the text (see p.6). We hope that the figure is now clearer.

(6) The "scaled I" network, known for representing input patterns in discrete attractors, should exhibit clear separation between network responses in the 2D PC space in the PCA plots. However, Figure 4D and Figure 6D do not reflect this, as all network responses are overlapped. Can the authors explain the overlap in Figure 4D?

In Figure 4D, activity of Scaled networks is distributed between three subregions in state space that are separated by the first 2 PCs. Two of them indeed correspond to attractor states representing the two learned odors while the third represents inputs that are not associated with these attractor states. To clarify this, please see also the density plot in Figure 4E. The few datapoints between these three subregions are likely outliers generated by the sequential change in inputs, as described in Supplementary Figure 8C.

(7) The reason for writing about the ISN networks is not clear. Co-tuned E-I assemblies do not necessarily have to operate in this regime. Also, the results of the paper do not rely on any of the properties of ISNs, but they are more general. Authors should either show the paradoxical effect associated with ISN (i.e., if increasing input to I neurons decreases their responses) or show ISN properties using stability analysis (See computational research conducted at the Allen Institute, namely Millman et al. 2020, eLife). Currently, the paper reads as if being in the ISN regime is a necessary requirement, which is not true. Also, the arguments do not connect with the rest of the paper and never show up again. Since we know it is not a requirement, there is no need to have those few sentences in the Results section. Also, the choice of alpha=5.0 is extreme, and therefore, it would help to judge the biological realism if the raster plots for Figs 2-6 are shown.

We have toned down the part on ISN and reduced it to one sentence for readers who might be interested in knowing whether activity is inhibition-stabilized or not. We have also added the reference to the Tsodyks et al. 1997 paper from which we derive our stability analysis. The text now reads 'Hence, pDp_sim_ entered a balanced state during odor stimulation (Figure 1D, E) with recurrent input dominating over afferent input, as observed in pDp (Rupprecht and Friedrich, 2018). Shuffling spike times of inhibitory neurons resulted in runaway activity with a probability of 0.79 ± 0.20, demonstrating that activity was inhibition-stabilized (Sadeh and Clopath, 2020b, Tsodyks et al., 1997).'

We have now also added the raster plots as suggested by the reviewer (see Figure 2D, Supplementary Figure 1 G, Supplementary Figure 4). We thank the reviewer for this comment.

(8) In the abstract, authors mention "fast pattern classification" and "continual learning," but in the paper, those issues have not been addressed. The study does not include any synaptic plasticity.

Concerning 'continual learning' we agree that we do not simulate the learning process itself. However, Figure 6 show results of a simulation where two additional patterns were stored in a network that already contained assemblies representing other odors. We consider this a crude way of exploring the end result of a 'continual learning' process. 'Fast pattern classification' is mentioned because activity in balanced networks can follow fluctuating inputs with high temporal resolution, while networks with stable attractor states tend to be slow. This is likely to account for the occurrence of hysteresis effects in *Scaled* but not *Tuned* networks as shown in Supplementary

Fig. 8.

(9) In the Introduction, the first sentence in the second paragraph reads: "... when neurons receive strong excitatory and inhibitory synaptic input ...". The word strong should be changed to "weak".Also, see the pioneering work of Brunel 2000.

In classical balanced networks, strong excitatory inputs are counterbalanced by strong inhibitory inputs, leading to a fluctuation-driven regime. We have added Brunel 2000.

(10) In the second paragraph of the introduction, the authors refer to studies about structural co-tuning (e.g., where "precise" synaptic balance is mentioned, and Vogels et al. 2011 should be cited there) and functional co-tuning (which is, in fact, different than tracking of excitation by inhibition, but the authors refer to that as co-tuning). It makes it easier to understand which studies talk about structural co-tuning and which ones are about functional co-tuning. The paper by Znamenski 2018, which showed both structural and functional tuning in experiments, is missing here.

We added the citation to the now published paper by Znamenskyi et al. (2024).

(11) The third paragraph in the Introduction misses some references that address network dynamics that are shaped by the inhibitory architecture in E/I assemblies in spiking networks, like Rost et al 2018 and Lagzi et al 2021.

These references have been added.

(12) The last sentence of the fourth paragraph in the Introduction implies that functional co-tuning is due to structural co-tuning, which is not necessarily true. While structural co-tuning results in functional co-tuning, functional co-tuning does not require structural co-tuning because it could arise from shared correlated input or heterogeneity in synaptic connections from E to I cells.

We generally agree with the reviewer, but we are uncertain which sentence the reviewer refers to.

We assume the reviewer refers to the last sentence of the second (rather than the fourth paragraph), which explicitly mentions the '…structural basis of E/I co-tuning…'. If so, we consider this sentence still correct because the 'structural basis' refers not specifically to E/I assemblies, but also includes any other connectivity that may produce co-tuning, including the connectivity underlying the alternative possibilities mentioned by the reviewer (shared correlated input or heterogeneity of synaptic connections).

(13) In order to ensure that the comparison between network dynamics is legit, authors should mention up front that for all networks, the average firing rates for the excitatory cells were kept at 1 Hz, and the background input was identical for all E and I cells across different networks.

We slightly revised the text to make this more clear 'We (…) uniformly scaled I-to-E connection weights by a factor of χ until E population firing rates in response to learned odors matched the corresponding firing rates in *rand* networks, i.e., 1 Hz'

(14) In the last paragraph on page 5, my understanding was that an individual odor could target different cells within an assembly in different trials to generate trial to trail variability. If this is correct, this needs to be mentioned clearly.

This is not correct, an odor consists of 150 activated mitral cells with defined firing rates. As now mentioned in the Methods, 'Spikes were then generated from a Poisson distribution, and this process was repeated to create trial-to-trial variability.'

(15) The last paragraph on page 6 mentions that the four OB activity patterns were uncorrelated but if they were designed as in Figure 4A, dues to the existing overlap between the patterns, they cannot be uncorrelated.

This appears to be a misunderstanding. We mention in the text (and show in Figure 4B) that the four odors which '… were assigned to the corners of a square…' are uncorrelated. The intermediate odors are of course not uncorrelated. We slightly modified the corresponding paragraph (now on page 7) to clarify this: 'The subspace consisted of a set of OB activity patterns representing four uncorrelated pure odors and mixtures of these pure odors. Pure odors were assigned to the corners of a square and mixtures were generated by selecting active mitral cells from each of the pure odors with probabilities depending on the relative distances from the corners (Figure 4A, Methods).'

(16) The notion of "learned" and "novel" odors may be misleading as there was no plasticity in the network to acquire an input representation. It would be beneficial for the authors to clarify that by "learned," they imply the presence of the corresponding E assembly for the odor in the network, with the input solely impacting that assembly. Conversely, for "novel" inputs, the input does not target a predefined assembly. In Figure 2 and Figure 4, it would be especially helpful to have the spiking raster plots of some sample E and I cells.

As suggested by the reviewer, we have modified the existing spiking raster plots in Figure 2, such that they include examples of responses to both learned and novel odors. We added spiking raster plots showing responses of I neurons to the same odors in Supplementary Figure 1F, as well as spiking raster plots of E neurons in Supplementary Figure 4A. To clarify the usage of 'learned' and 'novel', we have added a sentence in the Results section: 'We thus refer to an odor as 'learned' when a network contains a corresponding assembly, and as 'novel' when no such assembly is present.'.

(17) In the last paragraph of page 8, can the authors explain where the asymmetry comes from?

As mentioned in the text, the asymmetry comes from the difference in the covariance structure of different classes. To clarify, we have rephrased the sentence defining the Mahalanobis distance:

'This measure quantifies the distance between the pattern and the class center, taking into account covariation of neuronal activity within the class. In bidirectional comparisons between patterns from different classes, the mean dM may be asymmetric if neural covariance differs between classes.'

(18) The first paragraph of page 9: random networks are not expected to perform pattern classification, but just pattern representation. It would have been better if the authors compared Scaled I network with E/I co-tuned network. Regardless of the expected poorer performance of the E/I co-tuned networks, the result would have been interesting.

Please see our reply to the public review (reviewer 2).

(19) Second paragraph on page 9, the authors should provide statistical significance test analysis for the statement "... was significantly higher ...".

We have performed a Wilcoxon signed-rank test, and reported the p-value in the revised manuscript (p < 0.01).

(20) The last sentence in the first paragraph on page 11 is not clear. What do the authors mean by "linearize input-output functions", and how does it support their claim?

We have now amended this sentence to clarify what we mean: '…linearize the relationship between the mean input and output firing rates of neuronal populations…'.

(21) In the first sentence of the last paragraph on page 11, the authors mentioned 'high variability', but it is not clear compared with which of the other 3 networks they observed high variability.Structurally co-tuned E/I networks are expected to diminish network-level variability.

'High variability' refers to the variability of spike trains, which is now mentioned explicity in the text. We hope this more precise statement clarifies this point.

(22) Methods section, page 14: "firing rates decreased with a time constant of 1, 2 or 4 s". How did they decrease? Was it an implementation algorithm? The time scale of input presentation is 2 s and it overlaps with the decay time constant (particularly with the one with 4 s decrease).

Firing rates decreased exponentially. We have added this information in the Methods section.

**Reviewer #3 (Recommendations For The Authors):**
In the following, I suggest minor corrections to each section which I believe can improve the manuscript.- There was no github link to the code in the manuscript. The code should be made available with a link to github in the final manuscript.

The code can be found here: https://github.com/clairemb90/pDp-model. The link has been added in the Methods section.

Figure 1:- Fig. 1A: call it pDp not Dp. Please check if this name is consistent in every figure and the text.

Thank you for catching this. Now corrected in Figure 1, Figure 2 and in the text.

- The authors write: "Hence, pDpsim entered an inhibition-stabilized balanced state (Sadeh and Clopath, 2020b) during odor stimulation (Figure 1D, E)." and then later "Shuffling spike times of inhibitory neurons resulted in runaway activity with a probability of ~80%, demonstrating that activity was indeed inhibition-stabilized. These results were robust against parameter variations (Methods)." I would suggest moving the second sentence before the first sentence, because the fact that the network is in the ISN regime follows from the shuffled spike timing result.Also, I'd suggest showing this as a supplementary figure.

We thank the reviewer for this comment. We have removed 'inhibition-stabilized' in the first sentence as there is no strong evidence of this in Rupprecht and Friedrich, 2018. And removed 'indeed' in the second sentence. We also provided more detailed statistics. The text now reads 'Hence, pDpsim entered a balanced state during odor stimulation (Figure 1D, E) with recurrent input dominating over afferent input, as observed in pDp (Rupprecht and Friedrich, 2018). Shuffling spike times of inhibitory neurons resulted in runaway activity with a probability of 0.79 ± 0.20, demonstrating that activity was inhibition-stabilized (Sadeh and Clopath, 2020b).'

Figure 2:- "... Scaled I networks (Figure 2H." Missing)

Corrected.

- The authors write "Unlike in Scaled I networks, mean firing rates evoked by novel odors were indistinguishable from those evoked by learned odors and from mean firing rates in rand networks (Figure 2F)."Why is this something you want to see? Isn't it that novel stimuli usually lead to high responses? Eg in the paper Schulz et al., 2021 (eLife) which is also cited by the authors it is shown that novel responses have high onset firing rates. I suggest clarifying this (same in the context of Fig. 3C).

In Dp and piriform cortex, firing rates evoked by learned odors are not substantially different from firing rates evoked by novel odors. While small differences between responses to learned versus novel odors cannot be excluded, substantial learning-related differences in firing rates, as observed in other brain areas, have not been described in Dp or piriform cortex. We added references in the last paragraph of p.5. Note that the paper by Schulz et al. (2021) models a different type of circuit.

- Fig. 2B: Indicate in figure caption that this is the case "Scaled I"

This is not exactly the case 'Scaled I', as the parameter 𝝌𝝌 (increased I to E strength) is set to 1.

- Suppl Fig. 2I: Is E&F ever used in the manuscript? I couldn't find a reference. I suggest removing it if not needed.

Suppl. Fig 2I E&F is now Suppl Fig.1G&H. We now refer to it in the text: 'Activity of networks with E assemblies could not be stabilized around 1 Hz by increasing connectivity from subsets of I neurons receiving dense feed-forward input from activated mitral cells (Supplementary Figure 1GH; Sadeh and Clopath, 2020).'

Figure 3:- As mentioned in my comment in the public review section, I find the arguments about pattern completion a little bit confusing. For me it's not clear why an increase of output correlations over input correlations is considered "pattern completion" (this is not to say that I don't find the nonlinear increase of output correlations interesting). For me, to test pattern completion with second-order statistics one would need to do a similar separation as in Suppl Fig. 3, ie measuring the pairwise correlation at cells in the assembly L that get direct input from L OB with cells in the assembly L that do not get direct input from OB. If the pairwise correlations of assembly cells which do not get direct input from OB increase in correlations, I would consider this as pattern completion (similar to the argument that increase in firing rate in cells which are not directly driven by OB are considered a sign of pattern completion).Also, for me it now seems like that there are contradictory results, in Fig. 3 only Scaled I can lead to pattern completion while in the context of Suppl. Fig. 3 the authors write "We found that assemblies were recruited by partial inputs in all structured pDpsim networks (Scaled and Tuned) without a significant increase in the overall population activity (Supplementary Figure 3A)." I suggest clarifying what the authors exactly mean by pattern completion, why the increase of output correlations above input correlations can be considered as pattern completion, and why the results differs when looking at firing rates versus correlations.

Please see our reply to the public review (reviewer 3).

- I actually would suggest adding Suppl. Fig. 3 to the main figure. It shows a more intuitive form of pattern completion and in the text there is a lot of back and forth between Fig. 3 and Suppl. Fig. 3

We feel that the additional explanations and panels in Fig.3 should clarify this issue and therefore prefer to keep Supplementary Figure 3 as part of the Supplementary Figures for simplicity.

- In the whole section "We next explored effects of assemblies ... prevented strong recurrent amplification within E/I assemblies." the authors could provide a link to the respective panel in Fig. 2 after each statement. This would help the reader follow your arguments.

We thank the reviewer for pointing this out. The references to the appropriate panels have been added.

- Fig. 3A: I guess the x-axis has been shifted upwards? Should be at zero.

We have modified the x-axis to make it consistent with panels B and C.

- Fig. 3B: In the figure caption, the dotted line is described as the novel odor but it is actually the unit line. The dashed lines represent the reference to the novel odor.

Fixed.

- Fig. 3C: The " is missing for Pseudo-Assembly N

Fixed.

- "...or a learned odor into another learned odor." Have here a ref to the Supplementary Figure 3B.

Added.

Figure 4:- "This geometry was largely maintained in the output of rand networks, consistent with the notion that random networks tend to preserve similarity relationships between input patterns (Babadi and Sompolinsky, 2014; Marr, 1969; Schaffer et al., 2018; Wiechert et al., 2010)." I suggest adding here reference to Fig. 4D (left).

Added.

- Please add a definition of E/I assemblies. How do the authors define E/I assemblies? I think they consider both, Tuned I and Tuned E+I as E/I assemblies? In Suppl. Fig. 2I E it looks like tuned feedforward input is defined as E/I assemblies.

We thank the reviewer for pointing this out. E/I assemblies are groups of E and I neurons with enhanced connectivity. In other words, in E/I assemblies, connectivity is enhanced not only between subsets of E neurons, but also between these E neurons and a subset of I neurons. This is now clarified in the text: 'We first selected the 25 I neurons that received the largest number of connections from the 100 E neurons of an assembly. To generate E/I assemblies, the connectivity between these two sets of neurons was then enhanced by two procedures.'. We removed 'E/I assemblies' in Suppl. Fig.2, where the term was not used correctly, and apologize for the confusion.

- Suppl. Fig. 4: Could the authors please define what they mean by "Loadings"

The loadings indicate the contribution of each neuron to each principal component, see adjusted legend of Suppl. Fig. 4: 'G. Loading plot: contribution of neurons to the first two PCs of a *rand* and a *Tuned E+I* network (Figure 4D).'

- Fig. 4F: The authors might want to normalize the participation ratio by the number of neurons (see e.g. Dahmen et al., 2023 bioRxiv, "relative PR"), so the PR is bound between 0 and 1 and the dependence on N is removed.

We thank the reviewer for the suggestion, but we prefer to use the non-normalized PR as we find it more easily interpretable (e.g. number of attractor states in Scaled networks).

- Fig. 4G&H: as mentioned in the public review, I'd add the case of Scaled I to be able to compare it to the Tuned E+I case.

As already mentioned in the public review, we thank the reviewer for this suggestion, which we have implemented.

- Figure caption Fig. 4H "Similar results were obtained in the full-dimensional space." I suggest showing this as a supplemental panel.

Since this only adds little information, we have chosen not to include it as a supplemental panel to avoid overloading the paper with figures.

Figure 5:- As mentioned in the public review, I suggest that the authors add the Scaled I case to Fig. 5 (it's shown in all figures and also in Fig. 6 again). I guess for Scaled I the separation between L and M will be very good?

Please see our reply to the public review (reviewer 3).

- Fig. 5A&B: I am a bit confused about which neurons are drawn to calculate the Mahalanobis distance. In Fig. 5A, the schematic indicates that the vector B from which the neurons are drawn is distinct from the distribution Q. For the example of odor L, the distribution Q consists of pure odor L with odors that have little mixtures with the other odors. But the vector v for odor L seems to be drawn only from odors that have slightly higher mixtures (as shown in the schematic in Fig. 5A). Is there a reason to choose the vector v from different odors than the distribution Q?

The distribution Q and the vector v consist of activity patterns across the same neurons in response to different odors. The reason to choose a different odor for v was to avoid having this test datapoint being included in the distribution Q. We also wanted Q to be the same for all test datapoints.

What does "drawn from whole population" mean? Does this mean that the vectors are drawn from any neuron in pDp? If yes, then I don't understand how the authors can distinguish between different odors (L,M,O,N) on the y-axis. Or does "whole population" mean that the vector is drawn across all assemblies as shown in the schematic in Fig. 5A and the case "neurons drawn from (pseudo-) assembly" means that the authors choose only one specific assembly? In any case, the description here is a bit confusing, I think it would help the reader to clarify those terms better.

Yes, 'drawn from whole population' means that we randomly draw 80 neurons from the 4000 E neurons in pDp. The y-axis means that we use the activity patterns of these neurons evoked by one of the 4 odors (L, M, N, O) as reference. We have modified the Figure legend to clarify this: 'd_M_ was computed based on the activity patterns of 80 E neurons drawn from the four (pseudo-) assemblies (top) or from the whole population of 4000 E neurons (bottom). Average of 50 draws.'

- Suppl Fig. 5A: In the schematic the distance is called dE(Q¯,V¯) while the colorbar has dE(Q¯,Q¯) with the Qs in different color. The green Q should be a V.

We thank the reviewer for spotting this mistake, it is now fixed.

- Fig. 5: Could the authors comment on the fact that a random network seems to be very good in classifying patterns on it's own. Maybe in the Discussion?

The task shown in Figure 5 is a relatively easy one, a forced-choice between four classes which are uncorrelated. In Supplementary Figure 9, we now show classification for correlated classes, which is already much harder.

Figure 6:- Is the correlation induced by creating mixtures like in the other Figures? Please clarify how the correlations were induced.

We clarified this point in the Methods section: 'The pixel at each vertex corresponded to one pure odor with 150 activated and 75 inhibited mitral cells (…) and the remaining pixels corresponded to mixtures. In the case of correlated pure odors (Figure 6), adjacent pure odors shared half of their activated and half of their inhibited cells.'. An explicit reference to the Methods section has also been added to the figure legend.

- Fig. 6C (right): why don't we see the clear separation in PC space as shown in Fig. 4? Is this related to the existence of correlations? Please clarify.

Yes. The assemblies corresponding to the correlated odors X and Y overlap significantly, and therefore responses to these odors cannot be well separated, especially for Scaled networks. We added the overlap quantification in the Results section to make this clear. 'These two additional assemblies had on average 16% of neurons in common due to the similarity of the odors.'

- "Furthermore, in this regime of higher pattern similarity, dM was again increased upon learning, particularly between learned odors and reference classes representing other odors (not shown)." Please show this (maybe as a supplemental figure).

We now show the data in Supplementary Figure 9.

Discussion:- The authors write: "We found that transformations became more discrete map-like when amplification within assemblies was increased and precision of synaptic balance was reduced. Likewise, decreasing amplification in assemblies of Scaled networks changed transformations towards the intermediate behavior, albeit with broader firing rate distributions than in Tuned networks (not shown)."Where do I see the first point? I guess when I compare in Fig. 4D the case of Scaled I vs Tuned E+I, but the sentence above sounds like the authors showed this in a more step-wise way eg by changing the strength of \alpha or \beta (as defined in Fig. 1).Also I think if the authors want to make the point that decreasing amplification in assemblies changes transformation with a different rate distribution in scaled vs tuned networks, the authors should show it (eg adding a supplemental figure).

The first point is indeed supported by data from different figures. Please note that the revised manuscript now contains further simulations that reinforce this statement, particularly those shown in Supplementary Figure 6, and that this point is now discussed more extensively in the Discussion. We hope that these revisions clarify this general point.

The data showing effects of decreasing amplification in assemblies is now shown in Supplementary Figure 6 (Scaled[adjust])

- I suggest adding the citation Znamenskiy et al., 2024 (Neuron; https://doi.org/10.1016/j.neuron.2023.12.013), which shows that excitatory and inhibitory (PV) neurons with functional similarities are indeed strongly connected in mouse V1, suggesting the existence of E/I assembly structure also in mammals.

Done.